# In Mitosis You Are Not: The NIMA Family of Kinases in *Aspergillus*, Yeast, and Mammals

**DOI:** 10.3390/ijms23074041

**Published:** 2022-04-06

**Authors:** Scott Bachus, Drayson Graves, Lauren Fulham, Nikolas Akkerman, Caelan Stephanson, Jessica Shieh, Peter Pelka

**Affiliations:** 1Department of Microbiology, University of Manitoba, Winnipeg, MB R3T 2N2, Canada; bachuss@myumanitoba.ca (S.B.); gravesd@myumanitoba.ca (D.G.); fulhaml@myumanitoba.ca (L.F.); umakkern@myumanitoba.ca (N.A.); stephan4@myumanitoba.ca (C.S.); shiehj@myumanitoba.ca (J.S.); 2Department of Medical Microbiology and Infectious Diseases, University of Manitoba, Winnipeg, MB R3E 0J9, Canada

**Keywords:** NIMA kinases, Nek, mitosis, cell cycle, DNA damage, ciliogenesis

## Abstract

The Never in mitosis gene A (NIMA) family of serine/threonine kinases is a diverse group of protein kinases implicated in a wide variety of cellular processes, including cilia regulation, microtubule dynamics, mitotic processes, cell growth, and DNA damage response. The founding member of this family was initially identified in *Aspergillus* and was found to play important roles in mitosis and cell division. The yeast family has one member each, Fin1p in fission yeast and Kin3p in budding yeast, also with functions in mitotic processes, but, overall, these are poorly studied kinases. The mammalian family, the main focus of this review, consists of 11 members named Nek1 to Nek11. With the exception of a few members, the functions of the mammalian Neks are poorly understood but appear to be quite diverse. Like the prototypical NIMA, many members appear to play important roles in mitosis and meiosis, but their functions in the cell go well beyond these well-established activities. In this review, we explore the roles of fungal and mammalian NIMA kinases and highlight the most recent findings in the field.

## 1. Introduction and *Aspergillus* NIMA

The NIMA family of kinases is a relatively poorly understood family of serine/threonine kinases. The prototype, NIMA (NIMA—never in mitosis, gene A), was first identified in *Aspergillus* as a protein responsible for regulation of mitotic progression [1]. In this study, Morris screened temperature-sensitive mutants defective in nuclear division, septation, or distribution of nuclei along the mycelium and classified cell cycle mutants into two categories: those that arrested before entry into mitosis based on a co-ordinate drop in both spindle and chromosome mitotic indices are *nim* mutants for *n*ever *i*n *m*itosis, and those that arrested during mitosis based on a rise in the same indices are *bim* mutants for *b*locked *i*n *m*itosis. The *nim* mutants included a variety of cell cycle regulators, such as cyclin B, Cdc25, and DNA polymerase. The *bim* mutants included genes for spindle motors and components of the anaphase promoting complex. In his screens, Morris identified four alleles of a gene that he denoted *nimA* [2]. The *nimA* mutants were found to arrest in late G_2_ phase, with duplicated spindle pole bodies (the fungal equivalent of the centrosomes; [3]), indicating that NIMA not only affects chromosomal condensation but also other events of mitosis such as spindle formation. Soon thereafter, the gene for *nimA* was cloned and identified as a serine/threonine protein kinase called NIMA [4]. Overexpression of NIMA leads to a mitotic-like state, with chromosomes that have condensed and the appearance of aberrant spindles. Chromosomes were also observed to condense if cells were arrested in S phase prior to induction of overexpression of NIMA [5]. The activity of NIMA is tightly controlled by several different mechanisms that ensure the protein is active only during G_2_/M transition [4,6,7]. Mechanisms controlling the activity of the protein include transcriptional control of the *nimA* locus, phosphorylation at several different sites by Cdc2/Cyclin B complex, dephosphorylation by Cdc25, and loss of protein stability via proteolytic degradation.

Cdc2 activation and its nuclear localization is a prerequisite for entry into mitosis and, in most cases, once Cdc2 has been activated, mitosis will occur. This does not happen in *nimA*5 mutants [8]. In the *nimA5* mutant, Cdc2 is fully active but no mitotic entry occurs, suggesting that a functional NIMA is required for G_2_/M transition. Activation of NIMA is dependent upon the activity of Cdc2 [6], possibly being directly phosphorylated and activated by Cdc2. In *Aspergillus*, the nuclear membrane is not broken down during mitosis and Cyclin B/Cdc2 complexes are excluded from the nucleus in *nimA* mutants [9]. The NIMA protein is nuclear and is associated with chromosomes, spindles, and spindle pole bodies [10]. Cytoplasmic localization of Cdc2 will prevent proper activation of downstream Cdc2 targets that are nuclear. It is likely that NIMA promotes nuclear localization of the Cyclin B/Cdc2 complex, and this is supported by experimental evidence in which a mutation that suppresses the *nimA*1 allele was found to affect *sonA*, a homolog of the Gle2/Rae1 nuclear transporter [9]. Another mutation that suppresses *nimA*1 allele is *sonB*. The *sonB* gene encodes a homolog of the human NUP98/NUP96 precursor, which is cleaved to form a mature nucleoporin that contains the GLEBS domain that binds SONA [11]. NIMA and SONA/SONB interact and likely regulate the nuclear entry of mitotic factors, including NIMA itself, that promote entry and progression through mitosis in *Aspergillus*. Both SONA and SONB contain multiple NIMA consensus phosphorylation sites (FXXS/T) and it is likely that they are targets for phosphorylation, at least at some of these residues. Indeed, mitotic phosphorylation of other nucleoporins has been shown to play an important role in the dispersal of the nuclear pore complex (NPC) components [12]. NIMA has also been shown to phosphorylate histone H3 at serine 10 in *Aspergillus* during mitosis [10]. This phosphorylation is correlated with chromatin condensation. Furthermore, expression of NIMA in S phase leads to inappropriate histone H3 serine 10 phosphorylation, even in the absence of Cdc2 activity.

The NIMA protein is a 79 kDa polypeptide with an N-terminal kinase domain that defines the NIMA family of kinases. In addition to the conserved kinase domain, NIMA has a predicted coiled-coil motif that may be important for the oligomerization of the protein [13]. In addition to these motifs, NIMA has two PEST sequences that are involved in the ubiquitin-regulated proteolysis of the protein. The kinase domain is broadly conserved among different NIMA family members, and this sequence conservation is used to define whether a kinase belongs to the NIMA family. There are members of the NIMA family in other species that have the conserved kinase domain but lack some of the other features often associated with NIMA-related kinases, such as the coiled-coil region or the PEST motifs, or have acquired additional domains (for an overview of various members of the NIMA family of kinases, see [14]). It is reasonable to assume that, in these cases, the kinase domain has been co-opted for other purposes during the evolutionary history of an organism. This is supported by the large number of NIMA-related kinases that have been discovered in higher eukaryotes, such as in humans, which have 11 members.

More recent work has shown a genetic interaction between the Set1 methyltransferase, NIMA, and CDK1 [15]. The study shows that modifications of the Set1 residues on histone H4 phenocopied the lethality seen with compromised function of NIMA or CDK1. The role of NIMA in mitosis has been further expanded by showing that it interacts with TINA, a NIMA-interacting protein that localizes to spindle pole bodies and plays a role in negatively regulating astral pole bodies [16]. NIMA is required for the localization of TINA to the spindle pole bodies during initiation of mitosis [16]. Interestingly, TINA was found to associate with An-WDR8, a highly conserved WD40-domain protein with suspected functions in mitosis. Mitotic involvement of NIMA was solidified when it was shown that it was required for multiple events during mitosis, and not just entry, as was originally thought [17]. In this study, Govindaraghavan et al. showed that NIMA was required for proper formation of the bipolar spindle, nuclear pore complex breakdown, proper chromatin segregation, and nuclear envelope rearrangements required to form two nuclei at the end of mitosis. NIMA was also shown to be suppressed by SonC, a chromatin-associated factor involved in the DNA damage response (DDR) pathway [18]. It was proposed that SonC regulated NIMA activities in response to DNA damage and during mitotic events, potentially regulating large-scale chromatin states to ensure genome integrity. NIMA was shown to regulate the opening of septal pores, the connections between hyphae in filamentous fungi, in *Aspergillus* cells during interphase [19].

In interphase cells, it was shown that NIMA is localized to the open pores and remains there until it migrates to the nucleus prior to mitosis, which results in pore closure. Importantly, it was shown that inactivation of NIMA led to pore closure during interphase. Lastly, a study has implicated NIMA in cell polarization during *Aspergillus* growth [20]. These results highlight the multiple diverse functions of NIMA in *Aspergillus* and set the stage for greater diversity in more complex organisms.

*Aspergillus* NIMA was the prototypical member of the NIMA family and the founding member of this class of serine/threonine kinases. Its discovery and initial characterization as a key regulator of mitotic processes paved the way for future studies of NIMA-related kinases in higher eukaryotes.

## 2. NIMA-Related Kinases in Yeast 

*S. pombe* and *S. cerevisiae* each contain one member of the NIMA family of kinases, Fin1p in fission yeast and Kin3p in budding yeast [21,22]. Neither of these proteins is essential for growth in yeast, and, functionally, neither is able to rescue *Aspergillus nimA* mutants. Kin3p is not well characterized, but it appears to be functionally related to human Nek2, which will be discussed later. There are many parallels between the structure and function of *Aspergillus* NIMA and yeast Fin1p. One significant difference, however, is that NIMA is indispensable for G_2_/M progression and mitosis, whereas Fin1p is not. Additionally, Moura et al. [23] showed that Kin3p was required for proper DNA checkpoint arrest at G_2_/M following DNA damage. A growing body of evidence has also implicated several members of the mammalian NIMA family in playing a role in DNA checkpoint control; this will be discussed in greater detail later.

Fin1p is an 83 kDa protein, overexpression of which leads to premature chromatin condensation during any point in the cell cycle, independently of Cdc2 activity [21]. *fin1* null cells are viable but show an elongated morphology that is characteristic of cell cycle delay. The cell cycle delay occurs in G_2_; however, unlike *Aspergillus nimA* mutants, in yeast Cdc2 activation is delayed. Structurally, Fin1p is similar to NIMA, with an N-terminal kinase domain followed by a C-terminal coiled-coil motif and PEST sequences. Also, like NIMA, the levels and activity of Fin1p are tightly regulated, with maximal activity observed from metaphase–anaphase transition to G_1_ [24]. The protein localizes to the spindle pole bodies in late G_2_ and remains there for the duration of mitosis. Furthermore, *fin1* null cells show perturbations of the nuclear envelope, suggesting a role for Fin1p in regulating nuclear membrane dynamics during mitosis in yeast. Interestingly, this phenotype was greatly enhanced in the *cut11* null background. Cut11p plays a role in anchoring spindle pole bodies to the nuclear envelope during mitosis, and is also associated with the NPCs during interphase. Cut11p has two Fin1p consensus phosphorylation sites, however Fin1p does not phosphorylate Cut11p in vitro. Nevertheless, it is likely that Fin1p regulates Cut11p function in order to modulate nuclear membrane dynamics during mitosis. Another protein that has been found to be regulated by Fin1p is Plo1p [25], with Plo1p being the yeast homolog of Polo kinase (for a review of Polo kinases and their function, see [26]). Fin1p was found to regulate the association of Plo1p with spindle pole bodies during late G_2_ and mitosis [25], and overexpression of Fin1p caused premature recruitment of Plo1p to the spindle pole bodies.

The yeast homologues of NIMA have provided insight into the function of the protein, but with clear differences from *Aspergillus nimA*. Morphological differences, especially the lack of a nuclear envelope breakdown and associated mechanistic differences, in the nuclear structure during mitosis between higher eukaryotes and yeast also make it difficult to relate some of the observed effects to mammalian analogues.

## 3. Human NIMA-Related Kinases

There are 11 members of the NIMA family of kinases (Neks for NIMA rElated Kinase) in humans, named Nek1 through Nek11. The first mammalian NIMA-related protein kinase cloned was Nek1 [27], followed by subsequent identification of others, with the naming following the order of identification. All the Neks share a similarly conserved kinase domain that define them as NIMA-related kinases, and most (with the exception of Nek3, Nek5, and Nek11) contain a tyrosine inhibitory residue that protrudes into the catalytic site. Phylogenetic analysis of the full-length human Neks groups them into three distinct clades (Figure 1); clade 1 consisting of Nek4, 6, 7, 8, 9 and 10; clade 2 consisting of Nek11; and clade 3 consisting of Nek1, 2, 3, and 5. If we look only at the kinase domain, the phylogenetics change somewhat (Figure 2), but the number of clades remains the same. In this case, clade 1 consists of Nek6, 7, and 10; clade 2 consists of Nek1, 2, 3, 4, 5 and 11; and clade 3 consists of Nek8 and 9. Multiple sequence alignment also demonstrates that, although the majority of the Nek kinase domains are relatively similar, there are some outliers (Figure 2). Most notably, the kinase domain of Nek10 is significantly truncated versus the others by well over 50 amino acids. Other notable differences include small insertions found in Nek2, 10, and 11; as well as extended C-terminus residues found in Nek6 and 7. Interestingly, Nek10, lacking the C-terminus found within the kinase domains of other mammalian Neks, appears to be a dual-specificity kinase with serine/threonine and tyrosine phosphorylation activities [28]. The difference in the two alignment outcomes, either for the whole kinase or the kinase domain only, suggests different evolutionary constraints. It is likely that the kinase domain is more restricted in its ability to change and would result in a more conservative degree of evolutionary alterations. However, the whole protein is less restricted, so would have regions that evolve more rapidly to fit specific functional needs, which is not possible for the catalytically active domain that is restricted in evolution by its need to be enzymatically active.

Structural data on the various human Neks are scant, so we used artificial intelligence-based structural analysis tools to predict the potential structures of the human Nek kinase family (Figure 3). As the analysis shows, beyond the globular kinase domain, the various Neks differ significantly in structure. Notably, Nek8 and 9 have a characteristic seven-propeller Regulator of Chromosome Condensation 1 (RCC1)-like domain, which is located distinct from the kinase domain. It is hard to conclude how accurate the predicted structures are due to there being only three resolved Nek structures in the Protein Database (PDB); nevertheless, this provides a foundation for future structural analyses and may give valuable insights for mutagenesis studies despite certain oddities observed in some of the predicted structures. Importantly, the predicted structures show great diversity in how these proteins are shaped, reflective of their diverse functions within cells.

We have also collected subcellular localization data for each Nek available from The Human Protein Atlas (Figure 4). Most of the Neks showed at least partial cytoplasmic localization, with some showing more predominant nuclear staining. In particular, Nek7 and 11 showed strong nuclear staining with reduced cytoplasmic localization, while Nek2, 6, and 10 showed nucleocytoplasmic distribution; lastly, Nek1, 3, 4, and 9 showed predominantly cytoplasmic localization during interphase. We know from the literature that the distribution of these proteins changes with the cell cycle; therefore, only cells that are in interphase and are not actively dividing are shown for simplicity.

## 4. Mammalian Nek1

Nek1 was identified during a screen of mouse cDNA expression libraries with antibodies to phosphotyrosine [27]. Initially, Letwin et al. called Nek1 a dual-specificity kinase, phosphorylating serine/threonine residues as well as tyrosine. However, no other members of the NIMA family have been shown to be able to phosphorylate tyrosine residues, with the exception of a recent report showing tyrosine phosphorylation by Nek10 [28,37]. Among the Neks, Nek1 has been one of the most studied because of its association with numerous human diseases. Loss of function mutations of Nek1 have been identified in 3% of amyotrophic lateral sclerosis (ALS) patients [38,39]. Mutations in Nek1 resulting in ciliopathies cause short rib polydactyl syndromes (SRPS), and Nek1 overexpression has been found to be a potential oncotarget for various cancers (i.e., prostate, glioma) [40,41]. Moreover, Nek1 acts as a regulator of several cellular functions. It can be found in the basal body region distinct from α-tubulin, where it aids in the formation and stability of cilia [42,43]. The kinase regulates the progression of the cell cycle, has functions in gametogenesis, and plays a role in meiotic spindle formation [44]. More recently, researchers have concentrated on the role Nek1 plays in DDR and cell death, but the protein’s role in these processes is still poorly understood [45,46].

The predominant form of Nek1 consists of 1203 residues, with a predicted molecular mass of 137 kDa. Experimentally, the protein migrates at 180 kDa, with several smaller species observed [47]. Nek1 contains an N-terminal kinase catalytic domain of approximately 260 amino acids, a noncatalytic middle portion and C-terminal tail that consists of several predicted coiled-coil domains where dimerization and autophosphorylation occur [27]. The kinase also contains PEST sequences and a putative nuclear export signal (NES) implicated in regulated protein degradation and the induction of nuclear export, respectively [47]. 

Murine Nek1 is widely expressed throughout various tissues; however, the highest protein levels are observed in the testes, with lower levels present in the ovaries and other tissues. Therefore, it is not surprising that Nek1 has been implicated in spermatogenesis [48]. Subcellular distribution shows the protein to be primarily cytoplasmic and overexpression leads to chromatin condensation that does not correlate with histone H3 phosphorylation on Ser-10 [49]. Conversely, removal of cohesion along the chromosome arms in murine spermatocytes is dependent on the interaction between Nek1, wings apart-like homolog (WAPL), and protein phosphatase 1 gamma (PP1γ). During prophase 1 in meiosis, Nek1 regulates WAPL loading via phosphorylation of WAPL, mediated by PP1γ in order to promote loss of cohesion [50]. Nek1 has also been found to localize to the centrosomes in interphase, where it associates with mitotic spindle poles during mitosis and meiosis [44,51]. Furthermore, it was discovered that Nek1 regulates meiotic and mitotic spindle architecture and formation. During chromosome congression (the moving of chromosomes towards the spindle equator [52]) in meiosis, Nek1-deficient cells will form abnormal spindle organizations that will appear elongated or multipolar, resulting in abnormal congression [44]. Abnormal mitotic spindles occur in cells lacking Nek1 as it was found that Nek1 is required to interact with Adducin 1 (ADD1), a cytoskeletal protein, and Myosin X (MYO10), an atypical actin-based motor molecule of the myosin superfamily. ADD1 becomes hyperphosphorylated and deaminated without the presence of Nek1, resulting in its reduction and further inability to interact with MYO10, altering the profile and ratio on spindles [44]. Nek1-mutant mice suffer from dwarfism and shortened life span; the males that survive are sterile and females that survive are subfertile, the surviving mice suffer from anemia, show facial dysmorphism, and develop polycystic kidney disease (PKD) [49].

Yeast 2-hybrid screens have identified several proteins that interact with the human Nek1, among which were proteins identified to play a role in the development of PKD, double-stranded DNA break repair, and the G_2_/M transition [45]. Nek1 was found to be activated and upregulated due to DNA damage, especially in the presence of double-stranded DNA breaks (DSBs) [46,53,54,55]. Following DNA damage due to ionizing radiation (IR), the protein distribution was altered from predominantly cytoplasmic to distinct nuclear foci that also contained γ-H2AX staining and Mediator of DNA Damage Checkpoint 1 (NFBD1/MDC1), two proteins involved in the response to IR-induced DSBs [53,56]. Additionally, Nek1 has been found to form a complex with Cilia and Flagella Associated Protein 410 (CFAP410/C21ORF2) during homologous recombination (HR)-dependent DSB repair, where it is proposed that C21ORF2 may translocate Nek1 to the nucleus upon DNA damage [57]. In the presence of DNA damage induced by H_2_O_2_ or doxorubicin, Nek1 was also found to associate with the Tousled Like Kinase 1 (TLK1) and colocalize to the nuclei, where it likely forms a complex with γ-H2AX at the site of damage [56]. Recent studies have further implicated Nek1 in DDR. However, there is conflicting evidence regarding the role of Nek1 in checkpoint control with Ataxia Telangiectasia Mutated (ATM) and Ataxia Telangiectasia and Rad3-Related Protein (ATR). Chen et al. (2011) describe how Nek1 functions in checkpoint control that is independent of ATM or ATR activation as Nek1 was still found to be active and localized to sites of DNA damage even in the absence of activity by either upstream mediator kinases ATM or ATR [54]. However, in 2013 contradicting evidence emerged: it was found that Nek1 was required for ATR activation via its interaction with ATR-Interacting Protein (ATRIP) [58]. Nonetheless, it was established that the absence of functional Nek1 causes the failure of downstream ATM and ATR target kinases Checkpoint Kinases 1 and 2 (Chk1 and Chk2), and thus Nek1-deficient cells fail to arrest at mitotic and G_1_/S phase check points [54]. 

Nek1 also contributes to mitochondrial functions. Nek1 deficiency can alter mitochondrial functions as it is associated with phenotypes of reduced mitophagy, increased number of mitochondria, and increased mitochondrial reactive oxygen species (ROS) [59]. Studies found that the activation of Nek1 by TLK1 contributes to the stability of Voltage Dependent Anion Channel 1 (VDAC1) and thus to mitochondrial permeability and integrity [60]. Nek1 was also found to regulate apoptosis through this same mechanism. By directly phosphorylating VDAC1 and regulating this ion channel’s activity, Nek1 directly inhibits cell death [60,61]. These observations are highly intriguing, as the protein appears to both respond to DNA damage and suppress cell death, perhaps in order to give cells a chance to repair their DNA before deciding to proceed with the cell cycle or undergo apoptosis.

In addition to Nek1 playing a role in DDR and cell death, the kinase has been implicated in the function within cilia, which are thin cell projections. Nek1 is a regulator in nonmotile primary cilium formation [42,62]. The kinase interacts with Transcriptional Coactivator with PDZ Binding Motif (TAZ), an adaptor protein that plays a role in ubiquitin-mediated degradation of Polycystin 2 (PC2) [63]. Nek1 phosphorylates TAZ, which leads to enhanced ubiquitination of PC2 and its subsequent proteasomal degradation. Disruption of Nek1 or TAZ results in aberrant primary cilium formation and the development of PKD. This interaction with TAZ links Nek1 to the Hippo pathway, a trend we will see with other Neks later. Nek1 is also involved in ciliogenesis. The kinase competes for binding to Centrosomal Protein 104 (contains tubulin-binding TOG (for tumour overexpressed gene) domain and C2HC zinc finger array) with distal centriole-capping protein Centriolar Coiled-Coil Protein 110, which is anticipated to promote substrate phosphorylation that can further regulate cilia length [43]. Variants in Nek1 can lead to abnormalities in the cilia number and structure. In particular, SRPS patients have truncation variants, causing disruption of their fibroblasts [39,64].

Nek1, the initial human Nek to be identified, set the stage for further research into this extensive family of kinases. Its involvement in many cellular processes, including cell division, meiosis, mitochondria function, ciliogenesis, and the DDR pathway, pointed the way to what other Neks participate in. Additionally, Nek1 involvement in several human diseases have made this kinase a subject of much interest to researchers over the last decades.

## 5. Mammalian Nek2

Human Nek2 was first cloned in 1994 [65], along with human Nek3. Nek2 has the highest amino acid conservation with *Aspergillus* NIMA, including a C-terminal extension with a coiled-coil domain. Three splice variants for Nek2 are expressed, Nek2A, Nek2B, and Nek2C. The difference is due to alternative splicing that produces a larger protein (Nek2A) with a different C-terminus (Nek2B) [66]. Nek2C is similar to Nek2A but lacks an eight-amino-acid sequence within the C-terminus of Nek2A [67]. Unlike NIMA, Nek2 lacks PEST sequences. Instead, it has a KEN box and a D-box, which have been associated with ubiquitin-dependent protein degradation by the anaphase promoting complex (APC) [65,68]. Nek2A, but not Nek2B, has the APC destruction box and a protein phosphatase 1c binding site [69]. 

Nek2 shows a strong cell cycle-coupled pattern of expression. The protein is barely detectable during G_1_, with levels slowly rising during S phase and reaching the maximum at late G_2_. The kinase activity of Nek2 is connected to the cell cycle state of the cell; the protein has very low kinase activity during M and early G_1_, but high kinase activity during S and G_2_ [70]. Nek2A and Nek2B show a somewhat different expression pattern during the cell cycle [66], as well as differential subcellular localization. Nek2A is degraded in M phase, while Nek2B is degraded in G_1_. Within the cell, the different splice variants also show varying localization: Nek2A shows a uniform nucleocytoplasmic distribution, Nek2B shows a predominantly cytoplasmic localization, and Nek2C is primarily nuclear [67]. 

Nek2 was shown to phosphorylate several centrosomal proteins including C-Nap1, rootletin, and β-catenin [71,72,73]. C-Nap1 is a large centrosomal coiled-coil protein that plays a role in holding centrioles closely together. Phosphorylation of C-Nap1 and rootletin by Nek2 leads to the partial loss of these proteins from centrioles. Furthermore, active Nek2 induced loss or relaxation of the attachment between centrioles in interphase [74,75]. Nek2 was also shown to phosphorylate Ninein-like protein (Nlp), a centrosomal protein that is also phosphorylated by Plk1 (polo-like kinase 1), and, as a consequence, dissociates from γ-tubulin [76]. Phosphorylation of Nlp by Plk1 is enhanced through phosphorylation by Nek2. This parallels the function of yeast Fin1p, which recruits the yeast polo-like kinase, Plo1, to the spindle pole body [25]. Additionally, Nek2 shares a similar relationship with the microtubule-stabilizing protein centrobin, which is also phosphorylated by both Nek2 and Plk1, at different sites and to opposing effects—phosphorylation at the Nek2 site is antagonistic to these microtubule-stabilizing functions, while phosphorylation at the Plk1 site supports them [77]. Interestingly, regulation of Nek2A activity in controlling centriole splitting appears to be regulated by the components of the Hippo pathway [78]. The mammalian sterile 20-like kinase 2 (Mst2) and the scaffold protein Salvador (hSav1) were shown to directly interact with Nek2A and regulate its phosphorylation of C-Nap1 and rootletin at centrosomes. This study also demonstrated that, upon partial inhibition of the kinesin-5, Kinesin Family Member 11 (Kif11/Eg5), the hSav-Mst2-Nek2A pathway is critical for proper bipolar spindle formation [78]. Another study has demonstrated a relationship between Nek2 and centrosomal protein 68 (Cep68), a protein critical to centrosomal cohesion. The study showed that Nek2 phosphorylated Cep68, promoting its dissociation from the centrosome at the onset of mitosis and subsequent degradation via the Skip, Cullin, F-box containing (SCF) complex-mediated targeting to the proteasome [79]. Linking the mitotic kinases further, it was demonstrated that Plk1 functions upstream of Nek2 in regulating centrosome splitting [80]. Plk1 phosphorylates Mst2 in the Nek2–Mst2–PP1γ complex, inhibiting PP1γ recruitment and promoting centrosome disjunction. The presence of PP1γ in the Nek2 complex has an inhibitory effect on Nek2A kinase activity, antagonizing centriole separation. There also appears to be a role for Nek2 regulation of Erk2 localization to the centrosomes [81], with Nek2 kinase activity being essential for this function. Finally, downregulation of Nek2B levels, but not of Nek2A, leads to mitotic delay [82], and, upon mitotic exit, cells showed mitotic abnormalities (multinucleated cells).

Nek2 was also isolated as a protein interacting with Hec1 (for highly expressed in cancer 1; [83]). Nek2 phosphorylates Hec1 in vitro at a residue that has also been shown to be phosphorylated at G_2_/M transition in vivo [84]. Human Hec1 is required for proper recruitment of Monopolar Spindle 1 (MPS1) kinase and Mitotic Arrest Deficient (Mad1/Mad2) complexes to the kinetochore [85]. Hec1 depletion in HeLa cells resulted in spindle-checkpoint mediated arrest, while simultaneous depletion of Hec1 and Mad2 caused catastrophic mitotic exit. It has also been shown that Nek2 binds to and phosphorylates telomeric repeat binding factor 1 (TRF1), another protein involved with cell cycle regulation, at several locations, as a means of regulating mitotic regulatory functions of TRF1. This study also showed that Nek2 overexpression in breast cancer cells resulted in unaligned chromosomes during metaphase, but only in the presence of TRF1, implying that TRF1 is at least one of the means by which Nek2 overexpression leads to abnormal mitosis and chromosomal instability [86]. Nek2A was also reported to interact with Mad1 [87], and depletion of Nek2A caused chromosomal bridges, indicative of failure of the spindle assembly checkpoint. These observations suggest that Nek2 plays a role in control of the spindle checkpoint. Indeed, a study has shown that phosphorylation of Hec1 by Nek2 at S165 occurs specifically at misaligned kinetochores [88]. This phosphorylation appears to be critical for proper Mad1/Mad2 recruitment to misaligned chromosomes and spindle assembly checkpoint-initiated arrest.

There is increasing evidence suggesting that Nek2 might play a role in the DDR pathway [89]. DNA damage leads to reduced levels and activity of Nek2, which leads to the failure of centrosomes to separate and causes G_2_ delay. Inhibition of Nek2 activity following DNA damage appears to be mediated via its interaction with Protein Phosphatase 1 (PP1) [90]. Nek2 binds to both PP1α and PP1γ, and both appear to modulate Nek2 kinase activity [78,80,90]. Furthermore, in cells in which Nek2 has been knocked down using siRNA, there is impairment of centrosome separation and growth.

Nek2 has been shown to play a crucial role in vertebrate left–right asymmetry development, with a specific mutation having been linked to congenital heart disease. This regulation can be negated by any alteration in Nek2 expression, positive or negative. Knockdown leads to centriole defects at the left–right organizer, while overexpression results in premature cilia resorption, likely due to its interaction with Nup98, a nucleoprotein important for cilium resorption [91]. In addition, Nek2A kinase has been shown to be at least partially responsible for the displacement of distal appendages from the mother centriole, a process necessary for cilia resorption in the G_2_/M transition [92]. Finally, colocalization of Nek2 and Kif24 triggers phosphorylation of Kif24, which inhibits cilia formation in proliferating cells. Indeed, these proteins have been shown to be overexpressed in breast cancer cells, with their ablation restoring ciliation and reducing proliferation [93].

Based on its pivotal role in various systems involved in cellular replication, it comes as no surprise that many studies have shown Nek2 to be highly expressed in multiple cancers [94], often predicting a poor overall prognosis [95]. It is associated with poor prognosis [96,97,98], metastasis [99], drug resistance [100], and tumour recurrence [101,102] in hepatocellular carcinoma; loss of p53 heterozygosity in breast cancer cells [103]; and poor prognosis [104] and drug resistance in lung cancer [105], ovarian cancer [106], and myeloma [107,108,109]. As such, methods of Nek2 inhibition are a popular avenue of research for cancer therapy [110]. These include molecular agents such as imidazopyridine derivatives [111,112,113] and purine-based inhibitors [114], as well as upregulation of natural miRNA transcripts [115]. Additionally, anti-Nek2 siRNA is being investigated as a treatment option for colorectal cancer [116] and pancreatic cancer [117]. For a detailed review of Nek2 as a target in cancer therapy, see [118].

All told, Nek2 is a multifunctional protein with pivotal roles in development, DDR, and cell cycle regulation; its function in centriole dissociation serves as a common factor. As with many essential cell cycle regulators, it serves as a common hallmark of many cancers and shows promise as a treatment option in some. Without a doubt, further research into the roles of Nek2 will improve our understanding of the cell cycle, vertebrate development, and many types of cancer.

## 6. Mammalian Nek3

Human Nek3 was initially cloned together with Nek2 [65], and the murine homolog was cloned subsequently [119]. The human protein contains 459 residues and has the NIMA N-terminal kinase catalytic domain and a C-terminal coiled-coil motif. Since the initial discovery of Nek3, little has been learned about its function. Murine Nek3 differs in its expression pattern during the cell cycle from Nek2, showing marginal variation during the different phases of the cycle. Furthermore, antibody injections or overexpression of wild-type or kinase-inactive forms produced no observable phenotype. However, overexpression of a kinase-inactive Nek3 mutant leads to an increase in apoptosis in T47D cells [120]. Furthermore, these authors show that Nek3 interacts with Vav1 and Vav2 guanine nucleotide exchange factors (GEFs) for Rho GTPases Rac1, RhoA and Cdc42 in a prolactin-dependent manner. Stimulation of T47D cells with prolactin also induced Nek3 kinase activity and interaction of Vav2/Nek3 with the prolactin receptor. Activation of Rac1 was also shown to require Nek3 and Vav2. These observations suggest that Nek3 plays a role in a Vav2-mediated Rac1 signaling pathway in a prolactin-dependent manner. This was further supported by observations where Nek3 overexpression induced cytoskeletal changes in response to prolactin [121], whereas downregulation of Nek3 by siRNA reduced these changes. Reduction of Nek3 interfered with Rac1 activation and reduced cell migration and invasion of T47D cells. Some of these changes were likely due to an interaction between Nek3 and paxillin, which resulted in enhanced serine phosphorylation in paxillin, which likely contributed to some of the observed cytoskeletal changes [121].

Recently, Nek3 was reported to associate with Eps15 Homology Domain-Containing Protein 2 (EHD2) [122] but not to phosphorylate it. The consequences of this interaction are unclear at this point; however, EHD2 also associates with Vav1, another Nek3 target, phosphorylating it and regulating Vav1 GEF activity [122]. Interestingly, the interaction between Nek3 and EHD2 appears to be mediated by Vav1 and the complex may play a role in the reorganization of actin filaments at the sites of endocytosis in the plasma membrane. In line with these studies, Nek3 was found to regulate focal adhesion size and the reorganization of actin cytoskeleton into stress fibers [123]. This activity was modulated by the phosphorylation of Nek3 at the activation loop T165 by the extracellular signal-regulated kinase 1/2. This ultimately leads to enhanced breast cancer cell migration and metastasis, which was inhibited by a phosphorylation-deficient T165V mutant of Nek3. Additionally, Nek3 has been reported to play a role in the deacetylation of microtubules in neurons in an HDAC6-dependent manner [124]. Taken together, Nek3 appears to be an important player in the regulation of the actin cytoskeleton. Rho GTPases are important regulators of cytoskeleton function and are often deregulated in cancer [125], playing an important role in driving metastasis, so associating Nek3 with these proteins presents an attractive potential therapeutic target.

More recently, Nek3 has been implicated in cell cycle regulation and mitosis [126]. Specifically, Nek3 suppression is associated with chromosomal DNA bridge formation and an ensuing delay in cytokinesis. Nek3 silencing in HeLa cells led to an increase in the frequency of cells blocked in cytokinesis due to DNA bridges. Moreover, Nek3 was found to accumulate at the furrow of ingression during anaphase, as well as to localize with RhoA GTPase during cytokinesis [126]. Given that Rho GTPases function in cytokinesis by regulating the actin and myosin contractile ring that eventually forms the cleavage furrow, the functional interplay between Nek3 and RhoA GTPase suggests a role for Nek3 in cytokinesis. This observation parallels other Neks and may suggest a hitherto unknown role for Nek3 in preserving genomic integrity during mitosis.

Overall, Nek3 is a relatively poorly characterized member of the human NIMA family. Nevertheless, recent work has shown that it plays a role in several important cellular processes, including the regulation of the cytoskeleton and related cellular motility, cancer metastasis, and genomic integrity.

## 7. Mammalian Nek4

Nek4 is also known as Serine/Threonine Kinase 2 [127]. Human Nek4 is 841 residues and constitutes only the conserved kinase catalytic domain with no other domains characteristic of the other NIMA family of kinases [14]. Recent reports have implicated Nek4 as playing a role in cilia dynamics [128]. In ciliated cells, Nek4 was found to localize to basal bodies, and downregulation of the kinase by siRNA reduced the number of ciliated cells. Additionally, Nek4 status altered the sensitivity of cells to microtubule poisons, suggesting a role for the protein in microtubule regulation and hinting at a possible therapeutic target or predictive marker where microtubule poisons are used as standard treatment [129]. In recent years, Nek4 has been implicated in replicative senescence entry and DDR. Nek4 suppression was found to extend the number of population doublings required for entry into senescence in multiple human fibroblast cell lines [130]. When determining the mechanisms by which Nek4 suppression delayed entry into replicative senescence, Nguyen et al. [130] observed that Nek4-suppressed cells exhibited an increased proliferation rate and reduced transcription levels of the cyclin-dependent kinase inhibitor, p21. In addition, Nek4-suppressed cells showed defective cell cycle arrest upon etoposide-induced DNA damage. Further analysis revealed that this cell cycle arrest impairment was due to the interaction between Nek4 and DNA Protein Kinase catalytic subunit (DNA-PKcs), Ku70, and Ku80, the proteins that form the DNA–protein kinase complex involved in recognition and repair of DSBs via nonhomologous end joining [130]. Specifically, Nek4 suppression resulted in less overall binding of DNA-PKcs to DNA, reduced p53 activation, and decreased H2AX phosphorylation following etoposide-induced DNA damage. Considering that impaired DDR and extension of time to replicative senescence can lead to the accumulation of mutations and the expansion of malignant cells, a possible role of Nek4 in senescence is significant to our understanding of cancer. Following the discovery of a Nek4/DNA–PKcs interaction, Basei et al. identified yet another Nek4 interacting partner in the context of DDR [131]. Proliferating Cell Nuclear Antigen, a participant in DNA replication and the nucleotide excision repair pathway, was found to colocalize with Nek4 at damage sites in response to ultraviolet (UV) irradiation. Furthermore, a partial colocalization of Nek4 with Promyelocytic Leukemia Protein nuclear bodies, another cellular stress response structure, was also observed. The colocalization of Nek4 with multiple DNA repair proteins indicates a likely function of Nek4 in multiple DNA damage repair pathways.

In summary, and like Nek1 before it, Nek4 appears to play a role in cilia dynamics, the DDR pathway, is involved in regulation of cellular growth, and may play a role in other cellular stress responses. Much more work is needed to fully appreciate Nek4’s functions in the cell, but slowly they are being unraveled.

## 8. Mammalian Nek5

Human Nek5 is 889 residues and contains the N-terminal conserved catalytic domain as well as the C-terminal coiled-coil domain; it also contains a putative central helicase-like domain. Several pieces of evidence suggest that Nek5 may also participate in DDR. Firstly, Nek5 appears to promote centrosomal integrity and cohesion [132]. It was observed that the depletion of Nek5 led to a loss of pericentriolar material (PCM) components and reduced microtubule nucleation at centrosomes. During mitosis, Nek5-depleted cells showed delayed centrosomal separation and defective chromosomal segregation [132]. Secondly, when wild-type and Nek5-depleted cells were treated with etoposide to induce DNA damage, wild-type cells showed a significant increase in G_2_/M arrest, whereas Nek5-depleted cells continued through the cell cycle [133]. These observations imply that Nek5 may be involved in the G_2_/M checkpoint. In addition to altering cell cycle arrest in response to DNA damage, Nek5 has also been shown to attenuate etoposide-induced DNA damage. Whereas etoposide-induced DNA lesions were repaired after 6 h of recovery in control cells, some DNA damage was still observable after recovery in Nek5-depleted cells. When Nek5 was inducibly overexpressed in cells, less DNA damage was observed both immediately after etoposide treatment and after 6 h of recovery compared to the control. This suggests that the presence of Nek5 promotes rapid DDR and that the absence of Nek5 either increases DNA breaks, delays DNA repair, or both. Furthermore, the observed interaction between Nek5 and DNA Topoisomerase II Beta (TOPIIβ) following etoposide treatment points to a possible mechanism for Nek5-mediated DDR. After 5 min of recovery from etoposide treatment, Nek5/TOPIIβ interactions increased sevenfold compared to control cells before eventually returning to baseline interaction levels after 20 min of recovery [133]. Recent work has further implicated Nek5 in cancer processes. Specifically, Nek5 was shown to enhance breast cancer cell mesenchymal morphology and migration [134], and was also shown to enhance breast cancer cell proliferation via upregulation of Cyclin A2 expression, while downregulating Cyclin D1, D3, and E1 expression [135]. Lastly, Nek5 inhibited mitochondria-mediated cell death and respiration via interactions with mitochondrial proteins Cytochrome C Oxidase Copper Chaperone, Metaxin 2, and BCL2 Associated Transcription Factor 1 [136].

Nek5 is one of the least characterized of all the human Neks. Yet, it is clear that it functions in ways similar to its kin. Importantly, it plays a role in mitochondrial processes, genomic integrity, and mitotic processes, processes important to many human diseases. Future work is bound to decipher the mysteries of this kinase.

## 9. Mammalian Nek6 and Nek7

Mammalian Nek6 and Nek7 are closely related Neks [137] that share 77% sequence identity. Both of these Neks contain only the catalytic domain and no other known functional domains. As a consequence, both proteins are quite small. Nek6 is 313 residues, while Nek7, the smallest of the NIMA-related kinases, is 302 amino acids long [138]. Nek6 is a mostly globular, somewhat elongated protein with a disordered N-terminus, which it uses to interact with its cellular partners [139]. Because neither Nek6 nor Nek7 contain coiled-coil domains, it is unlikely that they are activated by trans-autophosphorylation, as this domain is required for the interaction of another Nek with Nek6, and subsequent Nek6 phosphorylation [140]. Rather, evidence suggests that Nek9 phosphorylates both Nek6 and Nek7 and regulates their function. Nek9 has been found to bind directly to Nek6 [140,141], and lead to phosphorylation of Ser-206 on Nek6 and the equivalent Ser-195 on Nek7, which leads to activation of both kinases in vitro. Furthermore, Nek9 is activated during mitosis at the same time as levels of Nek6 increase [141]. 

Nek7 retains an autoinhibited conformation, where a tyrosine side chain protrudes into the catalytic site, interacting with the activation loop. A mutation within this tyrosine residue results in a constitutively activated kinase. This conformation is only released after interaction with the noncatalytic domain of Nek9 [34,142]. Kinase activity of Nek6 and Nek7 was also found to be enhanced upon binding of the C-terminal region of Nek9, but this binding had no effect on mutants in the tyrosine residue [34]. This tyrosine downregulation motif has been found in all human Neks except for Nek3, Nek5, and Nek11. Once released from their autoinhibitory state and phosphorylated by Nek9, Nek6 and Nek7 are recruited to separate and distinct kinesins. Nek9 and Nek6 interact with the kinesin Kif20A, whereas Nek9 and Nek7 form a signaling module with Kif14. During anaphase, the Nek9–Nek6–Kif20A module localizes Kif20A to the central spindle, regulating its bundling activity. The Nek9–Nek7–Kif14 signaling module is also recruited to the central spindle for proper cleavage furrow ingression and abscission [143]. Nek6 also phosphorylates the Kif11 on Ser-1033, which promotes centrosome separation [144]. 

Nek9, Nek6, Nek7, and Nek2 are believed to play a role in in the establishment of the microtubule-based mitotic spindle [145]. Nek7 is reported as functioning in microtubule stability, where knockdown of Nek7 leads to substantially lower microtubule dynamics, while ectopic expression of Nek7 is reported to increase the rate of microtubule growth and instability [146]. Originally, it was thought that Nek6 and Nek7 were regulators of the p70 ribosomal S6 kinase [147]; subsequent results showed that this is not the case in vivo [140]. Maintaining spindle pole integrity and spindle morphology has been associated with the interaction of Nek7 with the Regulator of G protein signaling 2 (RGS2), causing RGS2 to localize to the mitotic spindle. When Nek7 is either overexpressed or knocked down, it causes reduction of γ-tubulin at the mitotic spindle pole [148]. Nek6 is also believed to function in regulating the mitotic spindle. Hsp72 is phosphorylated by Nek6 at Thr-66, allowing its association with the mitotic spindle via interaction with the proteins Cytoskeleton Associated Protein 5 and Transforming Acidic Coiled-Coil Containing Protein 3 [149,150]. This allows for stable K-fiber assembly. The phosphorylation of Hsp72 via Nek6 has also been found to promote centrosome clustering [150]. Nek6 has been described to localize to mitotic structures, particularly microtubules, while Nek7 localized to spindle poles [151]. Evidence for a role of Nek6, Nek7, and Nek9, in spindle formation comes from Rapley and Bertran, who showed that Nek6 interacts with the kinesin Kif11 [152,153]. Nek9 is phosphorylated by CDK1, allowing it to interact with Plk1, resulting in Nek9 activation, which in turn activates Nek6 and Nek7. Nek6 then phosphorylates a subset of Kif11 at Ser-1033 during mitosis, which, when combined with Kif11 phosphorylation by CDK1 at Thr-926, allows Kif11 to accumulate at the centrosome, enabling normal spindle formation and chromosome separation [154]. Nek7 has also been found to phosphorylate Kif11, causing the accumulation of microtubules and regulating dendrite growth [155]. Microtubule dynamics of interneurons is also believed to be attributed to Nek7, with knockdown causing axon development issues [156]. Nek6, Nek7, and Nek9 are all believed to be phosphor-motif amplifiers of Plk1 due to all having near-identical phosphorylation-site motifs to that of Plk1 [28]. In addition to regulating spindle dynamics, Nek7 has been reported to induce centriole duplication [157]. It was observed that depletion of Nek7 resulted in reduced centrosomal pericentriolar material (PCM) protein accumulation, while overexpression of Nek7 led to extra centrioles. A similar result was obtained when overexpressing Nek6, suggesting that, despite having distinct roles, some redundancy between Nek6 and Nek7 does exist.

During mitosis, Nek6 and Nek7 phosphorylate EMAP Like 4 (EML4) at Ser-144 and Ser-146, disrupting the binding of EML4 with the microtubule lattice. EML4 is believed to stabilize microtubules during interphase. Depletion of Nek9, the upstream activator of Nek6 and Nek7, or deletion of either Nek6 or Nek7, causes the increased affinity of EML4 for spindle microtubules, which has been found to interfere with chromosomal congression [158]. An EML4-ALK variant 3 fusion protein further recruits Nek9 and Nek7 to the microtubule to increase stability, which has been found to increase cellular migration [159].

Despite a great degree of similarity in sequence and structure, Nek6 and Nek7 appear to be regulated differentially during the cell cycle. Knockdown of Nek6 levels by siRNA or inhibition of its activity by overexpression of a kinase-inactive mutant leads to cell cycle arrest in M phase and apoptosis [160]. Mitotic arrest occurs in pro-metaphase and metaphase, respectively. The role of Nek7 in mitosis is less obvious [161]. Nek7 is essential for proper cell cycle progression. Depletion of Nek7 inhibits cells from progressing through G_1_ via downregulation of cyclins and CDKs [162]. Nek7 knockout has also been reported to cause chromosomal lagging, micronuclei formation, and cytokinesis failure in mouse embryonic fibroblast cells [163]. In hepatocellular carcinoma, it was observed that Nek7 kinase activity increased in response to serum starvation, while Nek6 activity and expression was decreased; however, Nek6 expression was then quickly restored after serum addition [164]. It is likely that the less conserved regions flanking the kinase domain play an important role in regulating the protein activity. These observations point to distinct roles in signaling pathways between the two closely related Neks. This is supported by siRNA knockdown of either kinase; knockdown of either Nek6 or Nek7 led to defective mitotic progression, indicating that the two kinases are not redundant and function independently [151]. Nek7 has also been found to interact with a distinct set of cellular proteins, including RGS2, Tubulin Beta Class I, MNAT1 Component of CDK Activating Kinase, and Pleckstrin Homology Domain Containing A8, that Nek6 does not interact with, strengthening the idea that, despite similarities, these kinases function independently of one another [148]. 

Nek6 has also been reported to interact with peptidyl-prolyl isomerase Pin1, an enzyme that isomerizes phosphorylated Ser/Thr-Pro bonds in proteins. Pin1 has been shown to interact with cell cycle regulators in a phosphorylation-dependent manner. Nek6 was found to associate with Pin1 in GST-pulldown assays as well as immunoprecipitations [165]; additionally, Nek6 and Pin1 mRNA levels were highly correlated in several hepatocarcinomas analyzed. 

Intriguingly, Nek6 was shown to phosphorylate the transcription factor Oct1 in its DNA binding domain [166]. The consequence of this phosphorylation appears to be Oct1 ubiquitination, which results in relocalization of Oct1 from mitotic chromatin to other mitotic structures where Oct1 appears to play an unknown role. 

Nek6 is believed to function in telomere length regulation via phosphorylation of the Tripeptidyl Peptidase 1 (TPP1), where Nek7 was not found to interact with TPP1 as strongly [167]. Telomere integrity is believed to be regulated in part by Nek7. Nek7 stabilizes the telomeric repeat binding factor 1 (TRF1), when it has been recruited to telomeres after DNA damage [168]. When TRF1 is phosphorylated by Nek7 at Ser-114, it is unable to interact with Fbx4, allowing TRF1 to evade ubiquitin tagging and eventual degradation. TRF1 is therefore able to form a Shelterin complex that shapes and protects human telomeres [169]. 

Nek7 is also involved in activation of the NLR Family Pyrin Domain Containing 3 (NLRP3) inflammasome and inflammatory disease [138]. When Nek7 is phosphorylated at Ser-204 by Polo-like kinase 4 (PLK4), Nek7 exhibits reduced affinity for NLRP3, an innate immune sensor. This in turn reduces the inflammasome activation via NLRP3 [170]. Nek7 further limits the NLRP3 inflammasome to interphase [171]. Nek7 and NLRP3 have both been found to have increased expression in response to ROS and are believed to function in inducing apoptosis [172]. 

Over the last few years, a clear role for Nek6 in carcinogenesis has been established. The first clue to this came several years ago when Nek6 was reported to be phosphorylated due to DNA damage [173]. DNA damage due to IR or UV has been found to cause the phosphorylation and inactivation of Nek6. Nek6 has also been found to be directly phosphorylated and inactivated by Chk1 and Chk2 members of the ATM/ATR-Chk1/Chk2 DNA damage checkpoint pathway. Importantly, it was shown that overexpression of Nek6 is able to override DNA damage induced G_2_/M cell cycle arrest [173]. Nek6 is overexpressed in malignant tumours in a number of human cancer cell lines and has been shown to play a role in causing distant metastasis [174,175,176,177,178,179], while ectopic expression of wild-type, but not kinase-dead, Nek6 promoted (while silencing of Nek6 decreased) anchorage-independent growth, a critical hallmark of cancer [180]. Until recently, it was not clear how Nek6 can promote cellular transformation, but recent reports have suggested that Nek6 is able to override p53-induced cellular senescence [181,182]. High levels of Nek6 appear to inhibit the cell cycle arrest-associated reduction in levels of cyclin B and CDK1 [182]. This may be due in part to the phosphorylation of the transcription factor STAT3 [183,184]. Under reduced serum conditions, knockdown of Nek6 induces premature cancer cell senescence [185]. Nek6 has also been found to cause Smad4 blocking, a member of the Transforming Growth Factor Beta (TGFβ) pathway. Binding of Smad4 to Nek6 blocks translocation of Smad4 to the nucleus, thus inhibiting the TGFβ/Smad signaling pathway [186]. 

Nek6 has also been found to play a role in the cytoskeletal gateway of drug resistance via directly interacting with Guanylate Binding Protein 1, creating drug-resistant cancers [180]. Overexpression of Nek6 has been linked to castration-resistant prostate cancer via the maintenance of certain interferon signaling genes [187]. The link between cancer and Neks is highly intriguing and presents a novel area for therapeutic development as kinases are readily susceptible to chemical inhibitors, and such inhibitors have previously been used as treatment options in the clinic [188,189,190]. Nek6 is also targeted and downregulated by the microRNA miR-141-3p [191], a tumour-inhibiting miRNA. Overexpression of Nek6 has been observed in clear cell renal carcinomas and is believed to be attributed to a reduction of miR-141-3p. 

Nek6 RNA is able to form circular RNAs (circRNAs), which is a covalently closed loop of RNA without either a 5′ cap or 3′ polyadenylated tail [192]. Overexpression of circNek6 has been found in thyroid cancer. Chen and colleagues believe that this form of circNek6 targets the microRNA miR-370-3p. Downregulation of miR-370-3p activates the Wnt signaling pathway [193], which plays a critical role in regulating cell fate [194]. Knockdown of Nek6 circRNA has been observed to cause inhibition of the expression of myosin heavy chain 9 (MYH9), allowing differentiated thyroid cancers to become more susceptible to iodine 121-based radiotherapy [195]. circNek6 has also been discovered to be overexpressed in non-small-cell lung cancer and is believed to sequester miR-832-5p, preventing its regulation of Breast Carcinoma Amplified Sequence 2 [196]. 

Like Nek6, Nek7 has been linked to several cancers due to its high levels of expression in them [197,198,199,200,201]. Nek7 is believed to be a promising biomarker or therapeutic target for treating various cancers [190,202,203,204]. Nek7 has, further, been found to be upregulated by the Hsp90 cochaperone UNC-45 myosin chaperone A in an effort to increase PCM organization and mitotic chromosomal alignment in tumours [205]. Wolf–Hirschhorn syndrome candidate 1, a histone methyltransferase that is overexpressed in squamous cell carcinoma, has been observed to regulate the expression of Nek7 [206]. These studies clearly highlight the links between various cancers and Nek6 and 7.

Despite Nek6 and 7 being the smallest of the neks and lacking all functional domains bar the catalytic domain, they still participate in a diverse set of activities. These activities include mitotic processes including cytokinesis, telomere maintenance, and cell cycle regulation. They are also thought to contribute to drug resistance in certain cancers.

## 10. Mammalian Nek8

Nek8 is another member of the NIMA family that is poorly characterized. The human Nek8 protein is 703 residues long and contains the N-terminal NIMA-like kinase domain as well as a central region that is similar to RCC1 [207]. RCC1 is the chromatin-bound guanine nucleotide exchange factor for the small nuclear GTPase Ran. RCC1 is important in coordinating the onset of mitosis with the S phase completion in mammalian cells (for a review of Ran and RCC1 functions, see [208]. Ran also functions in nucleocytoplasmic transport, with RanGTP being predominantly nuclear during the interphase. The Nek8 gene was originally identified and cloned in *jck* mice as responsible for polycystic kidney disease [207]. The disease-causing mutation was found to reside within the RCC1 motif, causing a change of glycine to valine. Overexpression of the mutant protein results in enlarged, multinucleated cells that have defects in actin cytoskeleton. Similarly, a mutation in the RCC1 domain of Nek8 leads to polycystic kidney disease in rats [209]. Likewise, injection of zebrafish embryos with morpholino antisense oligonucleotides corresponding to the zebrafish ortholog of Nek8 results in kidney cyst formation. Nek8 has also been found to be overexpressed in several different primary human breast carcinomas [210]. Interestingly, it was shown that Nek8 interacts with TAZ, linking Nek8 and kidney disease to the TAZ/Hippo signaling pathway, and potentially linking Nek8 to tumourigenesis [211]. This was supported by evidence from multiple organ dysplasia linking Nek8 mutations to an altered Hippo pathway and increased Myc levels [211]. Indeed, involvement of Nek8 in the Hippo pathway was further expanded when it was shown that Nek8 mutations contribute to renal cystic dysplasia via deregulation of YAP [212]. Further evidence has linked Nek8 with cancer more recently. A pseudogene that generates endogenous siRNAs, YPPM1K, was shown to target numerous cellular transcripts, including Nek8 [213]. Importantly, in some tumour samples, it was found that this pseudogene was transcribed at a reduced level, with a concomitant increase in Nek8 levels. It was also observed that overexpression of Nek8 led to the release of growth-inhibitory effects of YPPM1K. Strong evidence has linked Nek8 to DSBs and the activation of ATR-regulated replication stress [214]. In cells lacking Nek8, there was an increase in DSBs caused by replication stress, unscheduled origin firing, and fork collapse. The root cause of this was the loss of Nek8-mediated inhibition of cyclin A-associated CDKs. Significantly, the same mutations in Nek8 that cause renal ciliopathies were associated with enhanced DNA lesions, which disrupted the renal cell architecture. Nek8 was further implicated in DSB response when it was shown that it was required for RAD51 nuclear foci formation, following replication fork stalling that was driving DNA damage and DSBs [215].

Semi-quantitative RT-PCR of normal human tissue found Nek8 mRNA levels to be highest in the thyroid, adrenal glands, and skin, with lower levels present in the spleen, colon, and uterus. Overall, however, transcript levels were very low in all tissues analyzed. Cellular staining of a mouse kidney epithelial cell line showed the protein to be localized to the proximal region of the primary cilium, and the protein was undetectable in dividing cells [51].

Research has further confirmed that Nek8 plays an important role in cilia biology [216,217,218]. Nek8, which is mutated in the human cystic kidney disease nephronophthisis [219], was found to localize to the centrosome and proximal region of cilia in dividing and ciliated cells [218]. The localization of the kinase appears to be RCC1-domain-dependent; interestingly, Nek8 appears to be both activated (phosphorylated) and subsequently degraded following cell cycle exit and initiation of ciliogenesis. Mutations that cause nephronophthisis are present within the same region of the RCC1 domain of Nek8 as those that cause polycystic kidney disease in *jck* mice [219]. In *jck* mice, the mutant Nek8 protein is mislocalized on the cilia, showing localization along the entire length of cilia rather than the proximal side of primary cilia, as was observed for wild-type Nek8 [217]. The result of Nek8 mutation in *jck* mice is aberrant expression and localization of Polycystin 1 (PC1) and 2, which likely disrupts the cilia structure. Interestingly, Nek8 was reported to bind to PC2 but not PC1, causing enhanced phosphorylation of PC2, higher accumulation of both PC1 and PC2 at cilia, and abnormal cilia morphology [217]. PC2 is also regulated by Nek2, which suggests a possible interplay in the regulation of cilia between Nek2 and Nek8. Indeed, phenotypes of Nek8-null and *PC2*-null mice are reported to be strikingly similar [220]. These authors have also shown that, in Nek8-null mice, ciliogenesis was intact, but there was a misexpression of left-sided marker genes early in development, indicating that nodal ciliary signaling was disturbed. Further studies implicated Ankyrin Repeat and Sterile Alpha Motif Domain Containing 6 (ANKS6), a ciliary protein that was found to activate Nek8 via binding to its kinase domain [221]. The authors observed that Nek8 was required for proper targeting of ANKS6 to the ciliary inversin compartment. Mice with mutations in ANKS6 or Nek8 showed heterotaxy, cardiopulmonary malformations, and cystic kidneys. Interestingly, similar phenotypes were reported in two newborn brothers with mutations in Nek8 within the RCC1 domain, having presented with renal, cardiac, and hepatic anomalies [222]. Another study implicated the von Hippel–Lindau protein (pVHL) and hypoxia-inducible factors in the regulation of Nek8 expression, kidney function, and cilia maintenance [223]. It was reported that, in cells with wild-type pVHL, Nek8 expression was reduced as compared to cells that have lost pVHL, while hypoxia appeared to upregulate Nek8 expression via hypoxia response elements found in the promoter of Nek8 in a Hypoxia Inducible Factor (HIF) 1a- and HIF-2a-dependent manner.

Concluding, Nek8 and Nek9 form a unique clade of NIMA kinases in mammals due to the presence of the RCC1-like domain, with largely unknown functions. The roles of Nek8 in the cell are diverse, and include cilia regulation, processes deregulated in cancer, and DDR. Still, it is an enigmatic member of the family, and much more research is needed to fully appreciate its cellular roles.

## 11. Mammalian Nek9

Nek9 belongs to the same subgroup as Nek8; it has the central RCC1-like domain but is longer and contains a C-terminal coiled-coil region. The protein was initially identified based on its β-casein kinase activity isolated from the rabbit lung [224] (called Nek8 by these authors). Nek9 was also cloned as a protein associated with Nek6 ([141]; called Nercc1 by these authors). Human Nek9 is 979 amino acids long with an N-terminal kinase domain of the NIMA family, a central RCC1-like motif, and a C-terminal coiled-coil motif [224]). Nek9 mRNA is expressed widely, with the highest levels present in the heart, liver, kidneys, and testes [224]. Furthermore, Nek9 levels were found to be unchanged during the cell cycle. However, the protein was found to be phosphorylated during mitosis and nocodazole-induced mitotic arrest [141]. It is unclear what phosphorylates Nek9 during mitosis, but in vitro studies indicate that CDK1 can phosphorylate Nek9. When Nek9 was originally purified [224], it was found to be associated with a coiled-coil protein Bicd2, a human homologue of *Drosophila* Bicaudal D protein. *Drosophila* Bicaudal D associates with microtubules and its primary role, during development, is to ensure proper localization of developmental factors. The significance of the interaction between Nek9 and Bicd2 is still unclear. Human Bicd2 is involved in proper positioning of the centrosomes and the nucleus during G_2_ via the regulation of dynein and kinesin-2 function [225]. This observation may link Nek9 to regulation of nuclear and centrosomal positioning prior to entry of cells into mitosis.

Nek9 is localized to the cytoplasm and shows fine granular staining in a variety of different cell lines [141,226]. Despite having a nuclear localization signal, the protein shows only minimal nuclear staining, even when overexpressed [226]. Interestingly, it was observed that a kinase-inactive mutant of Nek9 showed a nucleocytoplasmic distribution, suggesting that active kinase is required for removal of the protein from the nucleus. Microinjections of anti-Nek9 antibodies during prophase lead to arrest in pro-metaphase. This suggests that the protein is involved in regulating chromosomal alignment during mitosis, probably by controlling spindle organization, since it was found to be associated with mitotic microtubules, and activated Nek9 (phosphorylated at the activation loop Thr-210) is concentrated at the centrosomes during mitosis [141,227]. Interestingly, knockdown of Nek9 via transient RNA interference did not lead to a phenotype in mammalian cells [226,227]. In contrast to transient RNA interference, stable RNA interference of Nek9 induced by expression of shRNA resulted in a prolonged G_1_ and S phases in HeLa cells [228]. In *Xenopus*, the Nek9 ortholog was found to associate with the γ-tubulin ring complex. Immunodepletion of *Xenopus* Nek9 from egg extracts results in delayed spindle assembly, reduction of bipolar spindles, and formation of abnormal microtubule structures; all of these defects are corrected by the addition of purified recombinant *Xenopus* Nek9. These results implicate Nek9 in the control of the microtubule organization. It is possible that phosphorylation of Nek6 (and possibly Nek7) by Nek9 [140,141] plays some role in these functions. Interestingly, knockdown of Nek6 leads to mitotic arrest in mammalian cells [151,160], but knockdown of Nek9 has no effect [226,227]. It is possible that Nek6 may be activated by another kinase in the absence of Nek9, perhaps the closely related Nek8. Nek9 was also reported to bind Ran [141], preferring RanGDP to RanGTP. This binding involves Nek9 catalytic domain as well as the RCC1-like domain. It has not been shown that Nek9 can induce nucleotide exchange on Ran. More recent work has shed further light on the functions of Nek9 in microtubule dynamics, albeit in *Xenopus*. A study has shown that Nek9 is capable of phosphorylating NEDD1, which is an adaptor protein important for centrosomal recruitment of γ-tubulin, which results in NEDD1 localizing to the centrosome together with this tubulin, further implicating Nek9 in centrosomal dynamics [229]. In mammals, Nek9 was shown to regulate spindle organization and cell cycle progression in mouse oocyte meiosis and in early embryonic mitoses [230]. The authors showed that Nek9 is recruited to the microtubule organizing center and may regulate microtubule nucleation and mitotic spindle organization, which agrees with other studies showing similar localization. Implications of Nek9 role in mitosis were further solidified when it was shown that the depletion of Nek9 results in mitotic catastrophe through impairment of mitotic checkpoint control and spindle dynamics [231]. More recently, it was demonstrated that recruitment of Nek9 to spindle poles required the expression of Axin1 in mouse oocytes [232], clearly implicating Nek9 in mitotic/meiotic spindle organization. Nek9’s role in mitosis goes beyond the organization of the microtubules as research has shown that Nek9 participates in two signaling cascades that control distinct kinesis during anaphase [143]. Nek9 was shown to be required for proper localization and bundling of Kif23 at the anaphase central spindle and, separately, Nek9 played a role in recruitment of the Citron Rho-Interacting Serine/Threonine Kinase to the anaphase midzone. A previous study found that Nek6 phosphorylates the kinesin Kif11 [152]; recent results have shed more light on the regulation of Kif11 phosphorylation by Nek6 and have clearly implicated Nek9 and Plk1 as upstream regulators of this process [153]. Plk1 directly phosphorylates and activates Nek9, and this phosphorylation is a prerequisite for prophase centrosome separation. Interestingly, regulation of Nek6 interaction with Nek9 appears to be negatively regulated by the interaction of Nek9 with Dynein Light Chain LC8-Type 1, a component of the dynein complex [233]. The net result of this interference is failure of Nek6 activation and inhibition of early centrosome separation. More recently, it was shown that Nek9 phosphorylates TPX2 Microtubule Nucleation Factor (TPX2), which then plays a role in Kif11-dependent centrosome separation prior to nuclear envelope breakdown [234]. These authors showed that Nek9 phosphorylates TPX2 NLS, preventing its nuclear import and enhancing its centrosomal localization, where it plays a critical role in microtubule aster formation and Kif11 localization.

Nek9 is highly expressed in tissues that do not have a high level of dividing cells (such as the heart), and is expressed at steady levels throughout the cell cycle. This suggests that the protein has functions outside of the cell cycle. Indeed, Nek9 was found to associate with FACT (for facilitates chromatin transcription) [228], which could point to a function of the protein during interphase. FACT is an abundant heterodimeric complex consisting of human Spt16 and SSRP1 proteins [235]. The Spt16/SSRP1 complex was originally discovered as a chromatin-specific elongation factor that was found to aid in the transcription of chromatin templates in vitro [236]. This heterodimeric protein complex functions by altering the nucleosome structure so that one histone H2A–H2B dimer is removed during enzyme passage [237]. FACT also possesses histone chaperone activity and can deposit core histones onto DNA. The functional relevance of the interaction between FACT and Nek9 is still unclear. It is possible that Nek9 plays a role in modulating chromatin structure, perhaps with the aid of its RCC1-like domain, during interphase.

Nek9 was found to be a target of adenovirus E1A oncoprotein [226]. E1A is the immediate-early gene first expressed during viral infection. It has a wide variety of functions (for a review, see [238]); importantly, it is critical for the induction of S-phase and modulates cellular and viral gene expression. Nek9 binds to E1A via a region closely mapping to the NLS and results in exclusion of the kinase from the nucleus [226]. This is a paradoxical result, in a sense, as E1A itself is predominantly nuclear, but suggests that the interaction may only occur within the nucleoplasm. Interestingly, when a deletion mutant lacking the RCC1-like domain of Nek9 is co-expressed with E1A, it is retained in the nucleus, suggesting that the RCC1-like domain may be important for the nuclear export of Nek9, possibly via its interaction with Ran. Nevertheless, these results suggest that E1A may be interfering with nuclear functions of Nek9, which may play roles in S-phase and/or cell cycle induction as well as regulation of gene expression. Later studies have shed further light on the interaction of Nek9 and E1A, as well as a new interaction between Nek9 and the Epstein–Barr Virus (EBV) tegument protein BGLF2 [239,240]. The interaction with E1A shows that Nek9 acts as a negative regulator of a subset of p53-activated genes. This study showed that Nek9 localizes to p53-regulated promoters and is able to suppress their activation during adenovirus infection. This study showed, for the first time, a role for Nek9 in regulating gene expression and p53 function [239]. Indeed, an earlier study showed the dependence of p53-mutated cells on Nek9, where depletion of Nek9 drove cellular senescence via upregulation of the cell cycle inhibitor p21 [241], whose promoter is bound by Nek9 [239]. A similar phenotype was observed with EBV BGLF2, where this viral protein upregulated p21 levels in a Nek9-dependent way, driving the G_1_/S cell cycle arrest required for efficient EBV replication [240]. Indeed, these findings point to a role for Nek9 in regulating the p53 tumour suppressor function, promoting cellular growth. Interestingly, previous studies have shown that Nek9 overexpression can induce aneuploidy [226], which contributes to oncogenic transformation. Indeed, a study [242] has shown that Nek9 is responsible for gemcitabine sensitivity in cancer cells, where depletion of this kinase sensitizes cells to replication stress and reduces recovery, clearly implicating Nek9 in cancer cell survival signals. These authors demonstrated that Nek9 levels increase in response to replication stress; Nek9 complexed with CHK1 and played a role in CHK1 activation in response to replication stress. Further work on a patient with mutant Nek9 implicated the kinase in cell cycle and ciliary defects, where patient-derived fibroblasts that lost expression of Nek9 were slow to proliferate, with reduced cell cycle progression, and showed a significant reduction in cilia formation [243]. Significantly, Nek9 was also implicated in the survival of glioblastoma cells, with worse outcomes with high levels of Nek9 expression [244]. Likewise, Nek9 was found to be upregulated in meningioma [245]. On the other hand, decreased Nek9 expression was correlated with aggressive breast cancer with poor prognosis [246]. It is possible that the cellular context of Nek9 expression may dictate patient outcomes, with some promoting longer survival and others having the opposite effect. Recent work has also implicated Nek9 in metastasis of gastric cancer via phosphorylation of ARHGEF2, Rho/Rac Guanine Nucleotide Exchange Factor 2 [247]. In essence, these authors showed that Nek9 can influence RhoA activation and enhance cellular motility in gastric cancer via this phosphorylation event.

Recent work has implicated Nek9 in autophagy [248]. In this study, the authors showed that Nek9 can suppress Microtubule-Associated Protein 1 Light Chain 3 Beta (MAP1LC3B)-mediated autophagy of p62/sequestosome 1 via phosphorylation of MAP1LC3B, an important component required for the formation of autophagosome. Additionally, a recent study [249] has shown that Nek9 can regulate ciliogenesis by acting as an autophagy adaptor for MYH9, a myosin heavy chain protein. Other recent work has also linked Nek9 to the Hippo pathway, like the closely related Nek8 [250]. A study showed that Nek9 can interact with the short form of the Prolactin Receptor (PRLR-SF) and acts as an intermediator between PRLR and Hippo signaling.

Nek9 has been extensively studied over the last two decades. The roles this kinase plays in the cell are diverse, and include regulation of a variety of mitotic processes, involvement in transcription, regulation of p53 function, and roles in viral infection, ciliogenesis, autophagy, and cancer. Together with Nek8, it is the other NIMA kinase with an RCC1-like domain, and like Nek8, the functions of this domain remain relatively mysterious.

## 12. Mammalian Nek10

Human Nek10 is one of the least studied Nek proteins to date. It is a relatively large protein, comprised of 1125 amino acids. Recent work has shown Nek10 to be a tyrosine/serine–threonine dual-specificity kinase [28], which makes it unique within the Nek family. Like other Neks, Nek10 shows a His–Arg–Asp motif within its catalytic domain, typical of kinases positively regulated via phosphorylation [145]. Research has demonstrated that Nek10 is able to autophosphorylate its tyrosine and serine residues [28]. Unlike the rest of the Nek family, however, it has a centrally located kinase domain, flanked by coiled-coil motifs, giving it a distinctive protein architecture. It harbours a PEST sequence in its C-terminus and four N-terminal armadillo repeats. Armadillo repeats have been reported to act as key regions for protein–protein interactions in other proteins; however, the exact function of the armadillo repeats in Nek10 remains unknown [139]. With respect to protein–protein interactions, there has been a scarcity of research regarding what exactly Nek10 interacts with. Recently, however, Nek10 has been implicated in several novel cellular interactions. These studies have indicated that Nek10 interacts with DDR proteins such as Structural Maintenance of Chromosomes 3, Ubiqutin C, ATRX, Protein Kinase DNA-activated Catalytic Subunit, and SUMO1, suggesting that Nek10 may have an important role in transcriptional regulation, DNA damage control, and chromosomal maintenance [251]. 

Haider et al. demonstrated the significance Nek10 has for the regulation of p53 transcriptional activity via phosphorylation of a p53 tyrosine residue [37]. This induces the concentrations of p21 in a Nek10-dependent manner [252]. Differences in p53 target gene expression primarily depend on the tyrosine kinase activity of Nek10. Serine phosphorylation-restricted mutants of Nek10 were unable to augment p53 target gene expression, therefore implying that the protein’s tyrosine kinase activity is of higher consequence with regards to p53 activation. This is similar to what has been reported for Nek9, which also plays a role in regulating the expression of p53-target genes [239]. This has potential implications in cancer as there has been recent research correlating Nek10 mutations and a variety of human cancers, with over 2.6% of cancers showing some Nek10 alteration [37]. In renal clear cell carcinoma, Nek10 is hemizygously deleted in 13% of cases [37]. This highlights the importance of continuing Nek10 research and expanding scientific knowledge on its roles within cancer cells.

Experiments have shown that the loss of Nek10 leads to an increase in cell proliferation and DNA replication, contrasting Nek6 and Nek7, where a loss of function results in a decrease in cell proliferation [37]. Studies demonstrated that the introduction of genotoxins, such as cisplatin, lessens Nek10 concentrations and impairs cellular p21 levels [37]. When IR damage is induced, concentrations of p21 increase in control cells with respect to Nek10 knockdown cells, suggesting that Nek10 plays a role in the maintenance of a DDR when exposed to environmental genotoxins [251]. Studies have also shown that Nek10 mediates G_2_/M cell cycle arrest following UV irradiation. Nek10 was found to be required for Erk1/2 pathway activation following UV-induced DNA damage, but not in response to stimulation with mitogenic stimuli, such as EGF. Experiments have demonstrated that the specificity of Erk1/2 activation in response to UV irradiation can be attributed to the ability of Nek10 to promote a noncanonical mechanism of MEK activation [253]. Furthermore, Nek10 was found to interact with Raf1 and MEK1, creating a ternary complex that results in autoactivation of MEK1 and subsequent Erk1/2 hyperactivation [37,253]. Following UV irradiation, the Nek10–Erk1/2 pathway was found to be essential for G_2_/M checkpoint maintenance [37]. Nek10 may encourage interactions with alternative regulators or allow for a permissive change in MEK, resulting in autoactivation [253]. These results are part of a growing body of evidence that suggests that some of the Neks play crucial roles in regulating cellular responses to DNA damage, and that Nek10 fills a critical niche within DDR pathways.

Recent research into Nek10 has revealed that it plays a role in many essential mitochondrial functions within human cells. Nek10 was confirmed to colocalize with Glutamate Dehydrogenase 1, a mitochondrial matrix enzyme that is responsible for the conversion of glutamate to alpha-ketoglutarate and ammonia, while concurrently reducing NAD(P)^+^ to NAD(P)H [254]. Moreover, Nek10-depleted cells show high concentrations of fragmented mitochondrial filaments, which is generally concomitant with impaired mitochondrial respiration and overall mitochondrial dysfunction. Silencing Nek10 expression was shown to increase mitochondrial damage, increase concentrations of ROS, and dampen rates of respiration [254]. 

Additionally, studies have shown that Nek10 is vital for ciliogenesis and the post-mitotic processes of cilia assembly, like several other Neks already discussed. In normal cilia, Nek10 expression is localized significantly and uniformly across the axoneme of the cilia [255]. As previously described, it has a key role in the G_2_/M checkpoint control, which is required for cilia assembly and biogenesis. The expression of kinase-dead mutants of Nek10 significantly reduced the number of ciliated cells. This has many implications in several human respiratory tract diseases revolving around mucociliary clearance [256].

Nek10 is the only NIMA kinase to have tyrosine phosphorylation activity in addition to serine/threonine phosphorylation, making it unique amongst this family. Similarly to other Neks, it is involved in a multitude of cellular processes, including DDR, chromosomal maintenance, transcription, mitochondrial functions, and ciliogenesis.

## 13. Mammalian Nek11

Human Nek11, first identified in 2002, is generally characterized as a DNA-damage response kinase and was originally thought to exist in only two alternatively spliced versions: Nek11L and Nek11S [257]. Nek11L is the larger of the two variants, with 645 residues, while Nek11S is composed of only 470 residues. The difference between the two is the length of their respective C-termini due to differential splicing. More recent, however, was the discovery of at least two more isoforms of Nek11: NekC (482 residues) and NekD (599 residues) [145]. The various isoforms seem to exhibit localization patterns of their own, with the longer variants (NekL and NekD) being predominantly cytoplasmic, and the shorter variants (NekS and NekC) showing a greater preference for the nucleus [258]. This suggests that the isoforms may play distinct roles within the cell, although how the roles vary is not well understood. Other localization studies have demonstrated that the nuclear-associated isoforms will localize to the nucleolus during interphase and are associated with the spindle microtubules during prometaphase and metaphase of mitosis [259]. Nek11L mRNA levels were found to increase during S phase and to peak at G_2_/M phases. Furthermore, following IR treatment, Nek11L was found to accumulate around areas of DNA damage, and was postulated to lend a hand in DNA repair. Data suggest Ku70 as a potential binding partner for Nek11. Ku70 is generally recruited to sites of DNA damage and will mediate repair through nonhomologous end joining. Further research analyzing Nek11’s role in DNA damage repair is needed to fully appreciate many of the aforementioned pathways it may participate in.

Experiments have shown Nek11 kinase activity to be elevated two-fold by DNA replication checkpoint arrest when induced by replication inhibitors aphidicolin, thymidine, and hydroxyurea. Nek11L was reported to be colocalized to Nek2A at the nucleoli, and has been found to interact with phosphorylated Nek2A, but not with its kinase-inactive mutant [259]. This complex formation was enhanced in G_1_/S-arrested cells, and this was associated with increased Nek2A autophosphorylation activity. Furthermore, Nek2A phosphorylates Nek11L at the noncatalytic C-terminus, a region that was shown to bind to the catalytic domain of Nek11L and repress its function. This phosphorylation releases the inhibitory effect of the C-terminus and leads to Nek11L activation. Whether or not the Nek2A/Nek11 complex is required to activate Nek11 for all its functions, such as in localization to DNA damage, is yet to be determined. Further studies have solidified the role Nek11 plays in the DDR pathway. Importantly, Melixetian et al. found that Nek11 regulated Cdc25a degradation via direct phosphorylation of a residue required for polyubiquitination and proteasomal degradation [260]. Significantly, activation of Nek11 was mediated by phosphorylation by CHK1. Similarly to Cdc25a, Nek11 appears to phosphorylate nuclear protein Bloom Syndrome Helicase (BLM), which is often recruited to sites of DNA damage [258]. This allows for the interaction of TopBP1, an important mediator protein in DNA replication checkpoint control [261]. These results uncover another layer of control of Cdc25a stability that relies on the coordinated function of CHK1 and Nek11 and demonstrates the array of protein interactions Nek11 is capable of.

Overexpression of the active form of Nek11L unexpectedly induced polyploidy within cells [262], similarly to what was previously observed for Nek9 overexpression [226]. This suggests that Nek11 activity may result in delayed cell cycle progression and cause endoreduplication [263] or may trigger the failure of effective cytokinesis. In either case, this finding is indicative of Nek11’s role within cell cycle control, with or without DNA damage present.

Nek11 research has been starting to focus on the relationship between Nek11 proteins and various cancers. In one study, Sabir et al. found that Nek11 isoforms were downregulated in four separate colorectal cancer cell lines. They determined that the upregulation of Nek11 resulted in destruction of Cdc25a. The downregulation of Nek11 could therefore be beneficial for tumour growth, as this would escalate Cdc25a concentrations within the cell and result in uncontrolled cell division [262]. In a separate paper, Nek11-depleted cells decreased in the G_2_/M phase after IR, suggesting that it may be partially p53-dependent. Furthermore, Nek11 depletion within various cell lines caused the induction of apoptosis and inhibition of long-term cell survival, therefore demonstrating that Nek11 is essential for normal cellular growth and function [258].

Taken together, Nek11 is a protein with several different isoforms identified, more than other Neks. It has a well-established role in the DDR pathway, potentially playing a role in nonhomologous end joining and cell cycle regulation via influences on stability of Cdc25a, potentially giving tumour cells a growth advantage.

## 14. Conclusions

The human NIMA-related family of kinases comprises 11 members, but to date we know very little about the function of many of them. Recent advances in our understanding of many of the members of this family have shed light on their function and importance. In particular, their role in cell cycle regulation, apparent involvement in cancer, and clear role in DDR make them potential therapeutic targets that may be impeded with small molecule inhibitors. Indeed, Neks have been implicated in many human diseases including developmental disorders, amyotrophic lateral sclerosis, polycystic kidney disease, lethal skeletal dysplasia, primary ciliary dyskinesia, nevus comedonicus, multiple cancers, and multiple organ dysplasia. It is clear, therefore, that the importance and implications of studying these kinases in relation to a variety of human disorders are significant. We still know very little about most members of this kinase family, but with continued efforts their secrets are slowly being elucidated.

## Figures and Tables

**Figure 1 ijms-23-04041-f001:**
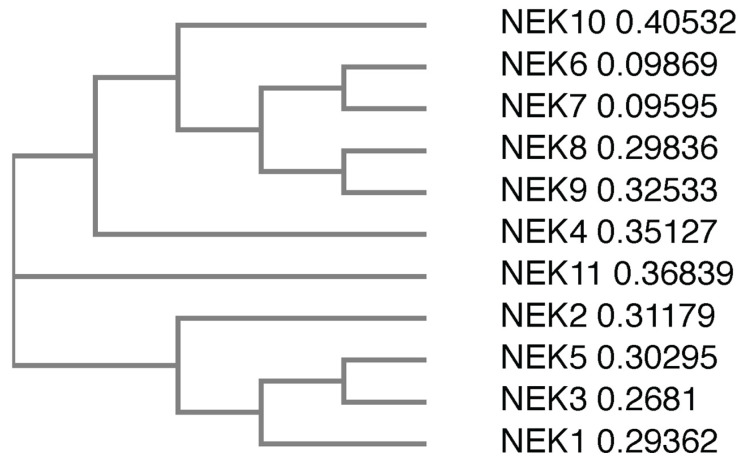
Phylogenetic tree of the full-length human Nek proteins. All full-length protein sequences were obtained from UniProtKB. Protein sequences were subsequently submitted to Clustal Omega, which generated a phylogenetic tree. The numbers next to the protein names refer to the sequence distance measure.

**Figure 2 ijms-23-04041-f002:**
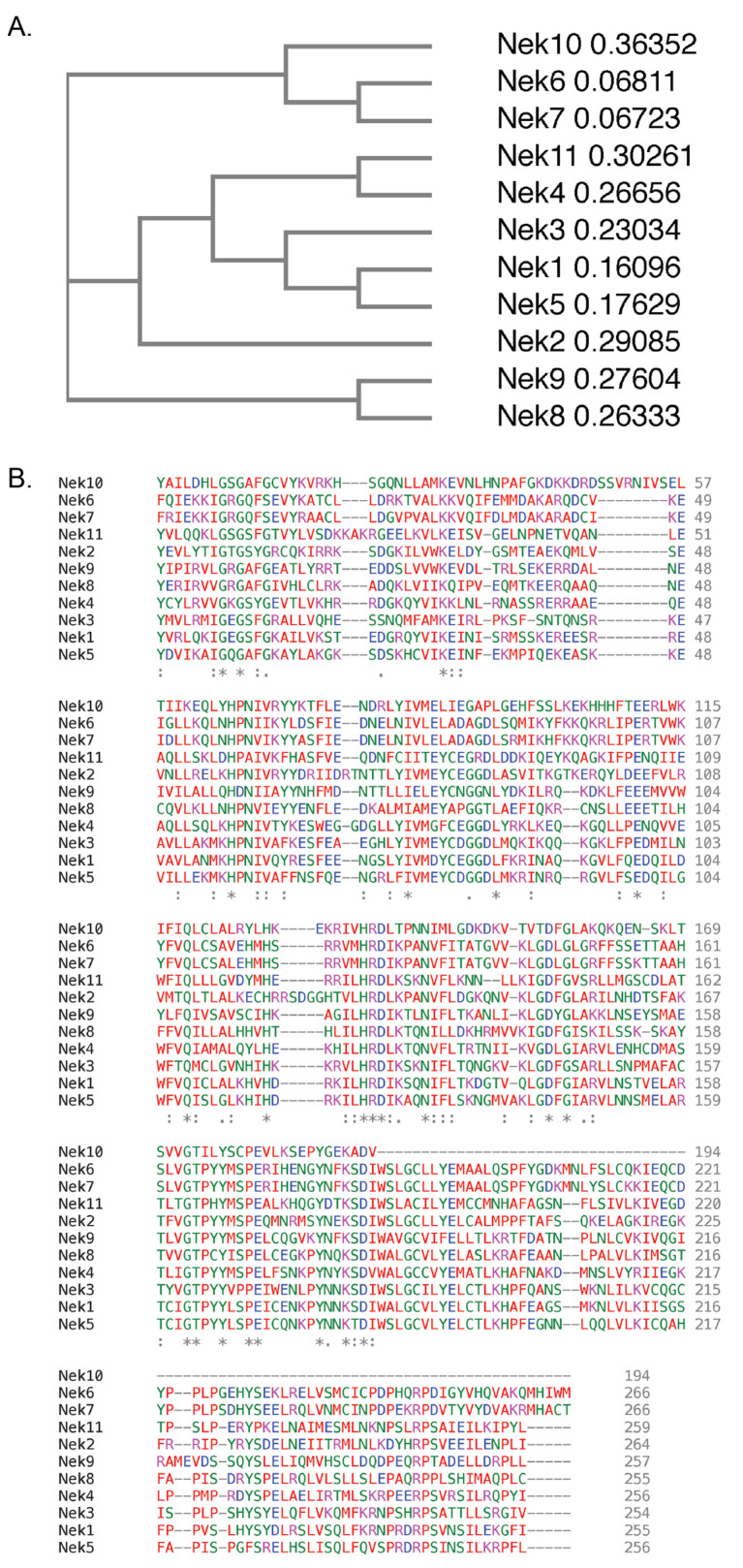
Phylogenetic tree and protein sequences alignment of the human Nek proteins kinase domains. All kinase domain protein sequences were obtained from UniProtKB. Protein sequences were subsequently submitted to Clustal Omega, which generated (**A**) a phylogenetic tree and (**B**) a multiple protein sequence alignment of the 11 human Nek protein kinase domains. Colours refer to amino acid properties as follows: red—small and hydrophobic including aromatic but excluding tyrosine; blue—acidic; magenta—basic excluding histidine; green—charged with hydroxyl, amine, or sulfhydryl groups, and glycine. * –fully conserved residues, : –conservation between groups of residues with strongly similar properties, . –conservation between groups of residues with weakly similar properties. The numbers next to the protein names in (**A**) refer to the sequence distance measure.

**Figure 3 ijms-23-04041-f003:**
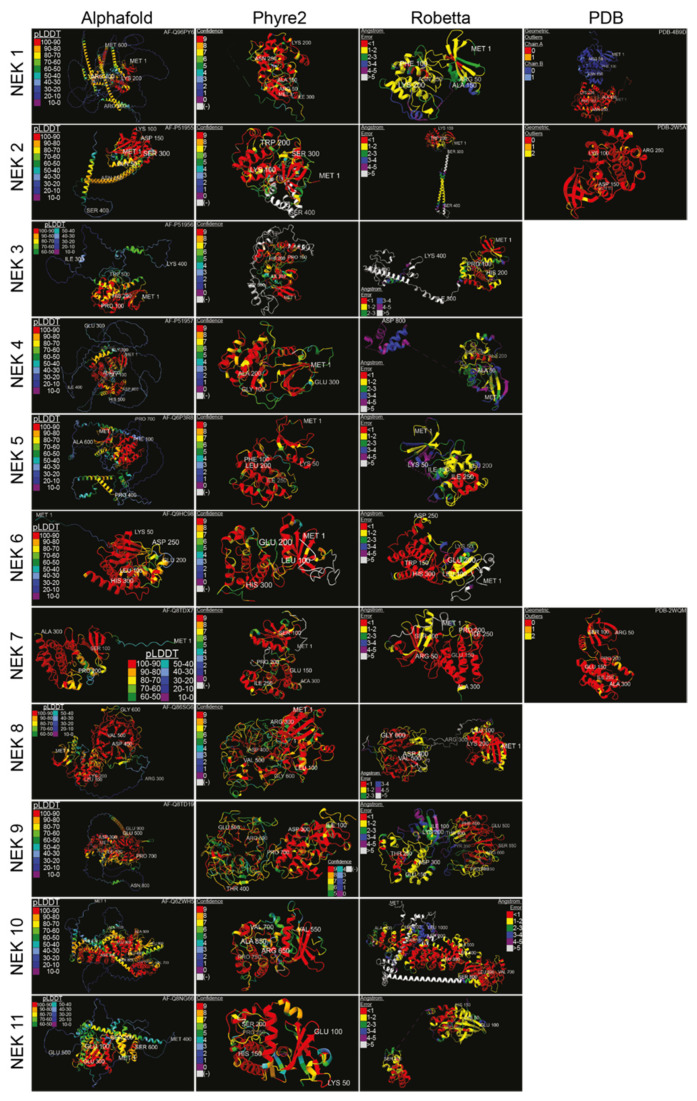
Structure predictions and known crystal structures of the human Nek proteins. Amino acid sequences were input into Robetta’s RoseTTAFold [29] and Phyre2′s Intensive [30] algorithms, with the resulting structures visualized in the first quarter of 2022. Amino acids were truncated from the C-terminus if necessary. Alphafold [31] and PDB structures were from [32] for Nek1 kinase domain, [33] for Nek2, and [34] for Nek7, with the accession numbers displayed. Structures were colour-coded according to the source’s measurement of error. High-error sequences were removed for visual clarity. Structures were visualized, colour-coded, and imaged with ChimeraX [35] (pLDDT = per-residue confidence score; (-) = outside of confidence range).

**Figure 4 ijms-23-04041-f004:**
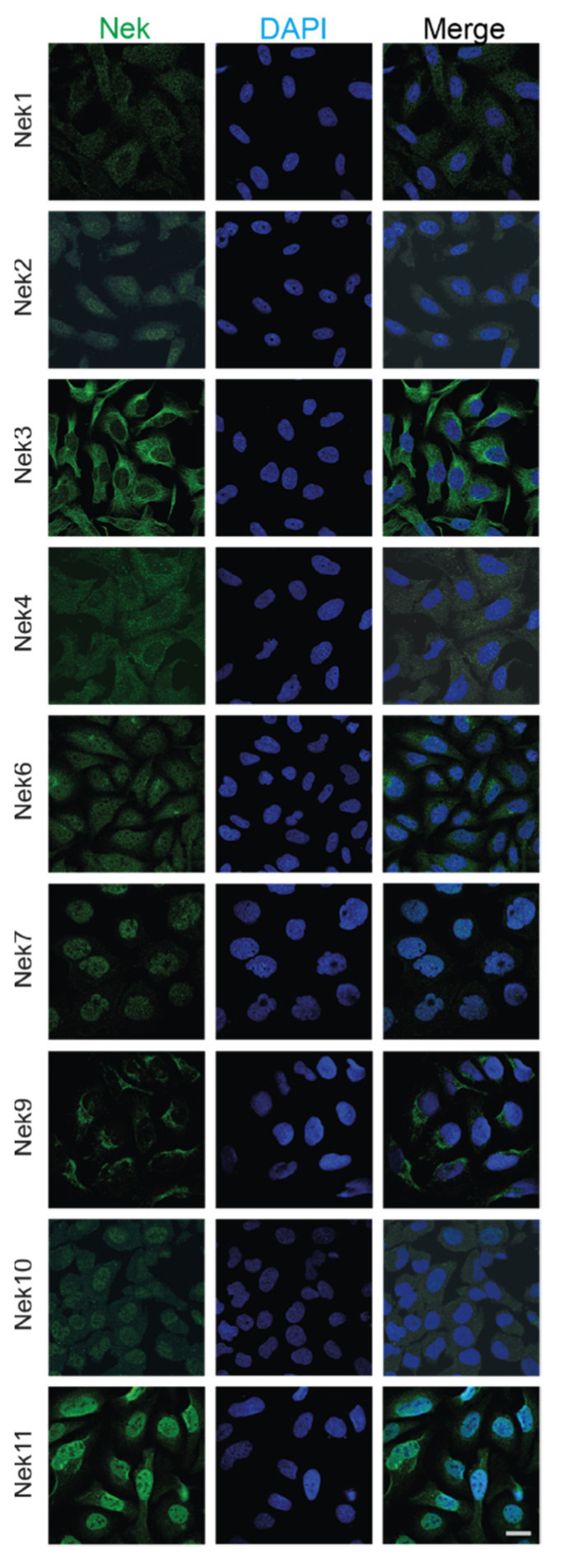
Subcellular localization of human Nek proteins in U2OS and A-431 cells. All images were obtained from the Human Protein Atlas and is reprinted with permission under Creative Commons Attribution-ShareAlike 3.0 International License from [36], copyright 2022 Human Protein Atlas, which did not have images for human Nek5 and Nek8; thus, they have been excluded. All images were taken in the human osteosarcoma cell line U2OS, except for Nek7, which was imaged in the human epidermoid carcinoma cell line A-431. The bar in the lowermost right image represents 20 μm and is the same for all panels.

## Data Availability

Not applicable.

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
