# Peer review of "In Mitosis You Are Not: The NIMA Family of Kinases in Aspergillus, Yeast, and Mammals"

_ijms, 2022, doi:10.3390/ijms23074041_

Round 1
Reviewer 1 Report
In this review, Bachus and colleagues describe the current literature on the NIMA-family of kinases in aspergillus, yeast and humans. First, they present an introduction to the discovery, and function of the founder member of the family, NIMA in aspergillus. Next, the authors present the latest literature on the function of NIMA-family members in yeast s. pombe and s. cerevisiae. Finally, the authors describe the function and relevance of the 11 mammalian Nek-kinases.
The review is comprehensive and, in the main, well written.
Broad comments:
One theme that runs through the piece, but which is particularly obvious in the section regarding mammalian Nek-kinases, is that the focus is on the detailed listing of individual experiments and observations, rather than providing a new comment or assessment on what these may tell us more broadly regarding the function, interactions and roles of these proteins. A summary to that effect at the beginning or end of each section would perhaps help steer the readers with a broader interest (rather than Nek-kinase specialists) to any general theme / salient details, while also maintaining some of the comprehensive details.
The figures provided with the text could also be improved to aid comprehension by amore generalist audience. For example, the alignments provided in figure 1A and 2, although interesting, don’t necessarily provide insight without a commentary on why the phylogenetic trees vary when comparing kinase domains and when comparing overall sequence. Similarly, the ‘take home message’ from figure 3 is not very clear, and it would be good if the features/residues/motifs that are common between NIMA-family members are highlighted.
The authors might consider including a ‘summary section’ at the end of the review, with a description of known pathologies and diseases linked to dysregulation of Nek kinases (all together in one section rather than dotted throughout the text). This would help a more general audience appreciate the importance and potential impact of the work on these proteins, and also provide a more obvious conclusion to the piece.
A figure summarising the architecture of protein domains in the different NIMA-family members might also be useful.
Specific comments:
Line 26: “Morris” is rather colloquial – reword to “In this study, Morris….”
Line 28: Please give more detail on how this phenotype is scored. – for clarity
Line 33: Explain, in slightly more detail, the relevance of the presence of duplicated SPB in this arrest.
Line 39 – Please give a little more detail. What kinases? How is the activity regulated (proximity to substrate / activity etc) ?
Line 46 (whole paragraph) – Please clarify / reword this sentence especially in line 48- the exclusion of cycB/cdc2 from the nucleus in a nimA mutant seems is a pretty good explanation for the arrest at G2/M phenotype.
Line 82: Reword “more advanced organisms” – perhaps to “higher eukaryotes”.
Line 87: “TINA” – Perhaps give a bit more details on this protein.
Line 91: Reword - “The study” perhaps to “In this study, Govindaraghavan et al.,”
Line 98: “Septal pores” Perhaps give a bit more of an explanation of what this is so it is more apparent to a non-specialist audience why this is intriguing.
Line 109: Reword “to rescue NimA mutants” perhaps to “ to rescue aspergillus NimA mutants”
Line 137: Reword “inappropriate” for clarity. Do you mean that Plo1 is recruited with different timings, to different levels, or to a different location?
Line 140: “morphological differences” – perhaps it would be nice to give a little more detail here for the non-specialist audience.
Comment – Would be nice – either in figure or table form to give a summary of the localisation and major mutant phenotype in both yeasts and aspergillus.
Line 144: It would be nice to know exactly what ‘Nek’ stands for.
Line 180: Reword – “Dearth” as it lack accuracy and is over-elaborate lag– are there any Nek kinase domains in the PDB at all?
Figure 4: Scale bars should be added to these figures.
Line 206: “These authors” – which authors are being referred to?
Line 208: Reword “to have tyrosine phosphorylation” perhaps to “to be able to phosphorylate tyrosine”
Line 212: Reword “Mutations in Nek1 involving ciliopathies” – do you mean “Mutations in Nek1 that result in ciliopathy”
Line 215: Reword: “Basal body region”. Is it possible to be more specific?
Line 217: Reword: “in meiotic spindles”. Is it possible to be more specific (for example is Nek1 found on the microtubules of the meiotic spindle?)
Line 219: Please give a reference for this possible involvement of Nek1 in the DDR and cell death.
Line 244: What sort of proteins are ADD1 and MYO10 ? Perhaps it would be nice to give a tiny bit more detail to help the reader understand the significance of this interaction.
Line 247: Reword – “those that survive are sterile males and subfertile females” – do you mean “the males that survive are sterile and females that survive are subfertile”?
Line 311: Reword – “degraded in the M phase” to “degraded in M phase”.
Line 320: Reword “complete splitting” for clarity. Do you mean that they are no longer attached and move apart?
Line 329: Reword “centriole dissociation” for clarity. Do you mean that they are moving apart from each other?
Line 338: Reword “degradation by” to perhaps something like “degradation via SCF mediated targeting to the proteasome”.
Line 339: Reword “centriole disjunction” for clarity. Do you mean that they are moving apart from each other?
Line 449: Reword “reduced cilia assembly”. Perhaps you could be a little more explicit, is the reduction in number, size, function etc?
Line 506: “Because neither Nek6 nor Nek7 contain coiled-coil domains it is unlikely that they are activated by trans-autophosphorylation”. Perhaps it would be nice to give a bit more detail here to explain the logic behind this statement.
Line 517: As the tyrosine downregulation motif is a common feature of most Nek kinases maybe it would be good to mention it earlier, possibly in section 3 where other common structural features are discussed.
Line 540: “Nek6 and Nek7 have been described to localize to spindle poles but with specific 540 differences”. Maybe it would be nice to very briefly give more details here about these different localisations.
Line 928: Change “ciliagenesis” to “ciliogenesis”
Line 933: Reword “This has many human implications for several respiratory tract diseases” to something like: “This has many implications in several human respiratory tract diseases”.
Author Response
One theme that runs through the piece, but which is particularly obvious in the section regarding mammalian Nek-kinases, is that the focus is on the detailed listing of individual experiments and observations, rather than providing a new comment or assessment on what these may tell us more broadly regarding the function, interactions and roles of these proteins. A summary to that effect at the beginning or end of each section would perhaps help steer the readers with a broader interest (rather than Nek-kinase specialists) to any general theme / salient details, while also maintaining some of the comprehensive details.
A brief summary, to each section, was added at the end of each as requested by the reviewer.
The figures provided with the text could also be improved to aid comprehension by amore generalist audience. For example, the alignments provided in figure 1A and 2, although interesting, don’t necessarily provide insight without a commentary on why the phylogenetic trees vary when comparing kinase domains and when comparing overall sequence. Similarly, the ‘take home message’ from figure 3 is not very clear, and it would be good if the features/residues/motifs that are common between NIMA-family members are highlighted.
We have added more context to the figure descriptions in the main text, for example we have expanded on the differences in the alignments and added why we think the structural pictures are informative.
The authors might consider including a ‘summary section’ at the end of the review, with a description of known pathologies and diseases linked to dysregulation of Nek kinases (all together in one section rather than dotted throughout the text). This would help a more general audience appreciate the importance and potential impact of the work on these proteins, and also provide a more obvious conclusion to the piece.
We have added this suggestion to the “Conclusions” section of the article. We feel this is appropriate and does not unnecessarily extend the length of an already lengthy review with a separate section.
A figure summarising the architecture of protein domains in the different NIMA-family members might also be useful.
We considered adding such a figure, but we ultimately decided against it as there are many such figures out there in other reviews and it would simply add to the length of an already long manuscript.
Line 26: “Morris” is rather colloquial – reword to “In this study, Morris….”
This was done.
Line 28: Please give more detail on how this phenotype is scored. – for clarity
This detail was added.
Line 33: Explain, in slightly more detail, the relevance of the presence of duplicated SPB in this arrest.
This was done.
Line 39 – Please give a little more detail. What kinases? How is the activity regulated (proximity to substrate / activity etc) ?
This detail was added.
Line 46 (whole paragraph) – Please clarify / reword this sentence especially in line 48- the exclusion of cycB/cdc2 from the nucleus in a nimA mutant seems is a pretty good explanation for the arrest at G2/M phenotype.
The phrasing of this section was changed to be more accurate as per reviewer’s suggestion.
Line 82: Reword “more advanced organisms” – perhaps to “higher eukaryotes”.
This was done.
Line 87: “TINA” – Perhaps give a bit more details on this protein.
More detail about TINA was added to the text.
Line 91: Reword - “The study” perhaps to “In this study, Govindaraghavan et al.,”
This was done.
Line 98: “Septal pores” Perhaps give a bit more of an explanation of what this is so it is more apparent to a non-specialist audience why this is intriguing.
An explanation of septal pores was added to the text.
Line 109: Reword “to rescue NimA mutants” perhaps to “ to rescue aspergillus NimA mutants”
This was reworded as per reviewer’s suggestion.
Line 137: Reword “inappropriate” for clarity. Do you mean that Plo1 is recruited with different timings, to different levels, or to a different location?
The wording was changed to provide an accurate description of what happens.
Line 140: “morphological differences” – perhaps it would be nice to give a little more detail here for the non-specialist audience.
More detail was added to the text as per reviewer’s suggestion.
Comment – Would be nice – either in figure or table form to give a summary of the localisation and major mutant phenotype in both yeasts and aspergillus.
We have considered this suggestion carefully, but since the focus of the review are mammalian kinases and it is already rather lengthy, we have decided not to add an additional table. Thank you for the suggestion.
Line 144: It would be nice to know exactly what ‘Nek’ stands for.
This was added.
Line 180: Reword – “Dearth” as it lack accuracy and is over-elaborate lag– are there any Nek kinase domains in the PDB at all?
The exact number was specified as per reviewer’s suggestion.
Figure 4: Scale bars should be added to these figures.
A scale bar was added to the figure.
Line 206: “These authors” – which authors are being referred to?
This was fixed to be specific.
Line 208: Reword “to have tyrosine phosphorylation” perhaps to “to be able to phosphorylate tyrosine”
This was done.
Line 212: Reword “Mutations in Nek1 involving ciliopathies” – do you mean “Mutations in Nek1 that result in ciliopathy”
This was done.
Line 215: Reword: “Basal body region”. Is it possible to be more specific?
Specific detail about the location was added to the text.
Line 217: Reword: “in meiotic spindles”. Is it possible to be more specific (for example is Nek1 found on the microtubules of the meiotic spindle?)
The appropriate detail was added to the text as suggested.
Line 219: Please give a reference for this possible involvement of Nek1 in the DDR and cell death.
The appropriate reference was added.
Line 244: What sort of proteins are ADD1 and MYO10 ? Perhaps it would be nice to give a tiny bit more detail to help the reader understand the significance of this interaction.
More detail about these proteins was added to the text.
Line 247: Reword – “those that survive are sterile males and subfertile females” – do you mean “the males that survive are sterile and females that survive are subfertile”?
This was done.
Line 311: Reword – “degraded in the M phase” to “degraded in M phase”.
This was done.
Line 320: Reword “complete splitting” for clarity. Do you mean that they are no longer attached and move apart?
This was clarified in the text.
Line 329: Reword “centriole dissociation” for clarity. Do you mean that they are moving apart from each other?
This was clarified in the text.
Line 338: Reword “degradation by” to perhaps something like “degradation via SCF mediated targeting to the proteasome”.
This was done.
Line 339: Reword “centriole disjunction” for clarity. Do you mean that they are moving apart from each other?
This was clarified now.
Line 449: Reword “reduced cilia assembly”. Perhaps you could be a little more explicit, is the reduction in number, size, function etc?
This was clarified in the text.
Line 506: “Because neither Nek6 nor Nek7 contain coiled-coil domains it is unlikely that they are activated by trans-autophosphorylation”. Perhaps it would be nice to give a bit more detail here to explain the logic behind this statement.
This statement was changed to be more accurate.
Line 517: As the tyrosine downregulation motif is a common feature of most Nek kinases maybe it would be good to mention it earlier, possibly in section 3 where other common structural features are discussed.
We have added this suggestion to section 3 of the article.
Line 540: “Nek6 and Nek7 have been described to localize to spindle poles but with specific 540 differences”. Maybe it would be nice to very briefly give more details here about these different localisations.
We have added more detail about the differences in the localization of these two Neks.
Line 928: Change “ciliagenesis” to “ciliogenesis”
This was done.
Line 933: Reword “This has many human implications for several respiratory tract diseases” to something like: “This has many implications in several human respiratory tract diseases”.
This was done.

Reviewer 2 Report
The authors have written a comprehensive and fact-based review on the NIMA-family of kinases. The organization and logic of the article are sound. There are no unjustified statements. The article merits publication and will make a contribution to the field as a solid current review of the facts.
However, before publication, there are a series of minor edits that should be made to improve the quality of the manuscript that I list below:
line 98: I suggest the authors start a new paragraph here, as the beginning of the paragraph is NIMA in mitosis, and then at this line the focus changes to the roles of NIMA in interphase
line 185: change "input" to the past tense "inputted"
Throughout the manuscript many reference brackets [###] are not separate by a space between them and the previous word - please correct the spacing format.
In addition to referencing the papers, it may be helpful to include a statement about which day/month/year the machine learning protein folding applications were used to create the modeled structures in Figure 3. This is not necessary, but may be helpful.
Line 198: please edit, "therefore this looks only at the cells that are in interphase and are not actively dividing"
A suggested edit is, "therefore only cells that were in interphase and were not actively dividing"
In Figure 4 the authors must include a scale bar.
Line 217: Use “and” not “thus”
Line 226: Edit "nuclear export induction" to “induction of nuclear export”
Line 229: “testis” should be the plural “testes” - please correct this throughout the manuscript
The authors sometimes write “subcellular” and at other times write “sub-cellular” - please be consistent. This is also true for “downregulation” and “down-regualtion”. Please check the manuscript carefully to be consistent.
Lines 247-248: "life span, those that survive are sterile males and subfertile females, suffer from anemia, show" edit to "life span, and those that survive are sterile males and subfertile females that suffer from anemia, show"
Line 252: "and G2/M transition" edit to "and the G2/M transition"
Please check the manuscript for the many, many undefined abbreviation terms. I do not list all of them here, but I use the following as an illustration - none of these terms were defined:
Line 256: gamma-H2AX
Line 256: NFBD1/MDC1
Line 257: C21ORF2
Line 260: TLK1
Line 263: ATM and ATR
Line 268: ATRIP
Line 276: VDAC1
Line 284: TAZ
Line 290: Cep104
Line 291: CP110
Line 260: H2O2 the "2" in this chemical formula looks like a small script font size, rather than a proper subscript. Please check this formatting.
Line 303: change "ubiquitin-mediated" to "ubiquitin-dependent"
Line 347: change "Hec1 (for highly expressed in cancer" to "Hec1 (for highly expressed in cancer 1"
The following 3 abbreviations (LR, LRO, and DAs) are only used this one time in the manuscript and seem unnecessary:
Line 372: vertebrate left-right asymmetry (LR) development
Line 375: left-right organizer (LRO)
Line 378: distal appendages (DAs) from the mother centriole
Line 733: "b-casein kinase" should use the Greek lowercase "beta"
Line 738: change "testis" to the plural "testes"
Line 774: change "shed the light" to "shed light"
In the manuscript, sometimes "et al." is in italics, and sometimes not - please be consistent
Words like "intriguingly" and "interestingly" are used, combined, more than 20 times - I think the use of these "attention-seeking" words should be more limited as a matter of style to really express what is most important in the author’s minds about the NIMA kinases. With this overuse, it is difficult for the the reader to judge if the authors are truly interested in the noted observation/function, or, if they simply find the observation/function as being unusual and notable, but not necessarily the most important function of the kinase.
Author Response
line 98: I suggest the authors start a new paragraph here, as the beginning of the paragraph is NIMA in mitosis, and then at this line the focus changes to the roles of NIMA in interphase
This was done.
line 185: change "input" to the past tense "inputted"
This was done.
Throughout the manuscript many reference brackets [###] are not separate by a space between them and the previous word - please correct the spacing format.
This was fixed.
In addition to referencing the papers, it may be helpful to include a statement about which day/month/year the machine learning protein folding applications were used to create the modeled structures in Figure 3. This is not necessary, but may be helpful.
This is specified in the figure legend now.
Line 198: please edit, "therefore this looks only at the cells that are in interphase and are not actively dividing" A suggested edit is, "therefore only cells that were in interphase and were not actively dividing"
This was reworded.
In Figure 4 the authors must include a scale bar.
A scale bar was added.
Line 217: Use “and” not “thus”
This was done.
Line 226: Edit "nuclear export induction" to “induction of nuclear export”
This was done.
Line 229: “testis” should be the plural “testes” - please correct this throughout the manuscript
This was changed throughout the manuscript.
The authors sometimes write “subcellular” and at other times write “sub-cellular” - please be consistent. This is also true for “downregulation” and “down-regualtion”. Please check the manuscript carefully to be consistent.
This was changed to be consistent throughout the manuscript.
Lines 247-248: "life span, those that survive are sterile males and subfertile females, suffer from anemia, show" edit to "life span, and those that survive are sterile males and subfertile females that suffer from anemia, show"
This was done.
Line 252: "and G2/M transition" edit to "and the G2/M transition"
This was done.
Please check the manuscript for the many, many undefined abbreviation terms. I do not list all of them here, but I use the following as an illustration - none of these terms were defined:
Line 256: gamma-H2AX
Line 256: NFBD1/MDC1
Line 257: C21ORF2
Line 260: TLK1
Line 263: ATM and ATR
Line 268: ATRIP
Line 276: VDAC1
Line 284: TAZ
Line 290: Cep104
Line 291: CP110
We have gone through the manuscript and changed the way abbreviations are used, adding definitions to those that are missing and removing abbreviations that were not referenced in the text after definition.
Line 260: H2O2 the "2" in this chemical formula looks like a small script font size, rather than a proper subscript. Please check this formatting.
This is a consequence of the font used and was indeed in proper subscript format.
Line 303: change "ubiquitin-mediated" to "ubiquitin-dependent"
This was done.
Line 347: change "Hec1 (for highly expressed in cancer" to "Hec1 (for highly expressed in cancer 1"
This was done.
The following 3 abbreviations (LR, LRO, and DAs) are only used this one time in the manuscript and seem unnecessary:
Line 372: vertebrate left-right asymmetry (LR) development
Line 375: left-right organizer (LRO)
Line 378: distal appendages (DAs) from the mother centriole
These were removed.
Line 733: "b-casein kinase" should use the Greek lowercase "beta"
This was done.
Line 738: change "testis" to the plural "testes"
This was done.
Line 774: change "shed the light" to "shed light"
This was done.
In the manuscript, sometimes "et al." is in italics, and sometimes not - please be consistent
This was changed to all italics to be consistent throughout.
Words like “intriguingly” and “interestingly” are used, combined, more than 20 times – I think the use of these “attention-seeking” words should be more limited as a matter of style to really express what is most important in the author’s minds about the NIMA kinases. With this overuse, it is difficult for the the reader to judge if the authors are truly interested in the noted observation/function, or, if they simply find the observation/function as being unusual and notable, but not necessarily the most important function of the kinase.
We have gone through the manuscript and reduced these words to be more selective.
